# Nonzero Berry phase in quantum oscillations from giant Rashba-type spin splitting in LaTiO$_3$/SrTiO$_3$ heterostructures

M.J. Veit[1], R. Arras[2], B.J. Ramshaw[3,4], R. Pentcheva[5] & Y. Suzuki[1]

The manipulation of the spin degrees of freedom in a solid has been of fundamental and technological interest recently for developing high-speed, low-power computational devices. There has been much work focused on developing highly spin-polarized materials and understanding their behavior when incorporated into so-called spintronic devices. These devices usually require spin splitting with magnetic fields. However, there is another promising strategy to achieve spin splitting using spatial symmetry breaking without the use of a magnetic field, known as Rashba-type splitting. Here we report evidence for a giant Rashba-type splitting at the interface of LaTiO$_3$ and SrTiO$_3$. Analysis of the magnetotransport reveals anisotropic magnetoresistance, weak anti-localization and quantum oscillation behavior consistent with a large Rashba-type splitting. It is surprising to find a large Rashba-type splitting in 3$d$ transition metal oxide-based systems such as the LaTiO$_3$/SrTiO$_3$ interface, but it is promising for the development of a new kind of oxide-based spintronics.

[1] Department of Applied Physics and Geballe Laboratory for Advanced Materials, Stanford University, Stanford, CA 94305, USA. [2] CEMES, University of Toulouse, CNRS, UPS, 29, rue Jeanne Marvig, 31055 Toulouse, France. [3] Los Alamos National Laboratory, Los Alamos, NM 87545, USA. [4] Laboratory for Atomic and Solid State Physics Cornell University Ithaca, NY 14853, USA. [5] Department of Physics and Center for Nanointegration (CENIDE), University of Duisburg-Essen, Lotharstrasse 1, 47057 Duisburg, Germany. Correspondence and requests for materials should be addressed to M.J.V. (email: mjveit@stanford.edu)

In the presence of an electric field, inversion symmetry in a solid is broken which, in addition to spin–orbit coupling, lifts the spin degeneracy[1]. This effect is known as Rashba-type spin splitting, and the resulting Fermi surface consists of two pockets, an inner Fermi surface and an outer Fermi surface, which have a helical spin texture. While the Rashba effect-inducing electric field is usually externally applied, such an electric field can also be produced at an interface with a polar discontinuity[2]. In complex oxide interface systems such as LaTiO$_3$/SrTiO$_3$ (LTO/STO)[3] and LaAlO$_3$/SrTiO$_3$ (LAO/STO)[4], the asymmetry of the potential in a direction perpendicular to the interface gives rise to Rashba-type spin–orbit interactions whose magnitude can be modulated by the application of an external electric field[5,6].

Bulk LTO is a Mott insulator on the verge of a metal–insulator transition with a small band gap of 0.1 eV[7]. Metallicity in LTO can be induced by cation vacancies, excess oxygen, alkaline-earth doping of the rare-earth site, and epitaxial strain[8–14]. Furthermore, it has been shown that LTO juxtaposed with STO induces metallicity[15,16] and even superconductivity[17] at the interface, and it has been proposed that the system orders magnetically[18,19].

LTO/STO heterostructures have previously been compared to the LAO/STO system, but there are a few differences which make these systems distinct from each other. In particular, when LTO is grown under large (−1.6%) compressive epitaxial strain on STO substrates, the films are metallic throughout the bulk of the film[15]. This metallicity is due to the fact that the epitaxial strain induces an electronic structure modification[20]. However, there is also a contribution to conduction from electronic reconstruction at the LTO/STO interface which has a higher resistivity[16]. While the LAO/STO system has been extensively studied for a wide range of LAO thicknesses, the magnetotransport of LTO/STO has been largely studied in films with thicknesses on the order of or >10 unit cells. This includes electrical gating dependence of the electrical transport and analysis of the magnetoconductivity in the framework of weak anti-localization[21–24]. However, in this thickness regime, our previous studies[15] have demonstrated that the electrical transport is dominated by the metallic conduction inside the entire LTO film induced by the epitaxial strain. We have therefore chosen to study ultra-thin (3–4 unit cells) films of LTO grown on STO single crystal substrates. These ultra-thin films exhibit properties drastically different from thicker films including previously unobserved quantum oscillations and a nonlinear Hall effect under zero bias[21].

In our study, we have observed two sets of Shubnikov–de Haas (SdH) oscillations, substantial anisotropic magnetoresistance (AMR) and a weak anti-localization correction to the magnetoconductivity in LTO/STO heterostructures—all of which are consistent with strong Rashba-type spin–orbit coupling. One set of SdH oscillations is observed at relatively low fields before reaching the quantum limit and a Berry phase of $\pi$ is observed in these oscillations. To our knowledge, this is the first such observation in an oxide heterostructure. Another set of oscillations is observed at higher field. These two sets of oscillations enable us to deduce a Rashba-type coupling coefficient of $1.8 \times 10^{-11}$ eVm which is an order of magnitude larger than that observed in LAO/STO. The magnetoconductivity exhibits a field dependence that can be attributed to weak anti-localization. This weak anti-localization correction to the magnetoconductivity provides an estimate for the spin–orbit relaxation time from which a similar Rashba-type coupling coefficient of $2. \times 10^{-11}$ eVm is deduced. Additionally, a large in-plane AMR much larger than other oxide systems is observed and is consistent with a Rashba system.

## Results

**SdH oscillations.** Electronic transport measurements of ultra-thin LTO films (3–4 unit cells thick) on STO substrates exhibit quantum oscillations characteristic of a clean system. Such oscillations have never before been seen either in LTO films or at the LTO/STO interface. Samples thicker than ~2 nm do not show any oscillations. Additionally, the ultra-thin samples show a significantly higher resistivity than the thicker samples (see Supplementary Note 1), indicating that there are two thickness regimes where different conduction mechanisms dominate. In the thick (>2 nm) regime, tetragonal distortions modify the electronic structure giving rise to metallicity in the bulk of the LTO. However, for ultra-thin (3–4 unit cells) films electronic reconstruction at the interface dominates conduction. The oscillations are therefore a measurement of the electronically reconstructed interface.

Figure 1 shows the isothermal longitudinal magnetoresistivity and Hall resistivity of a 1.2-nm-thick film with fields up to 9 T applied perpendicular to the film. Oscillations in the longitudinal magnetoresistance can clearly be seen in the raw data at fields between approximately 1 and 3 T but are emphasized in the derivative (shown in the inset). The oscillations quickly disappear above ~3 T. These oscillations are also seen in the Hall effect measurements. Figure 2 also shows longitudinal magnetoresistance for the same film up to 60 T. These measurements show a second set of different oscillations.

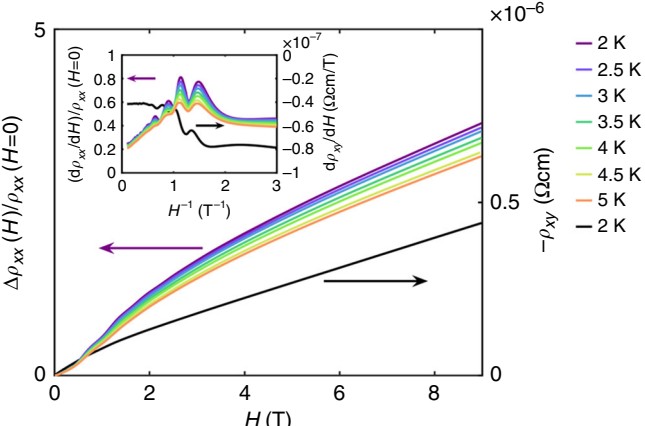

**Fig. 1** Low field Shubnikov–de Haas oscillations. Isothermal magnetoresistivity curves (left axis) and Hall resistivity (right axis) of a 1.2-nm-thick LTO film on STO at 2–5 K and 2 K, respectively. Oscillations are clearly visible between 1 and 3 T. The inset shows the derivative to highlight the oscillations even further

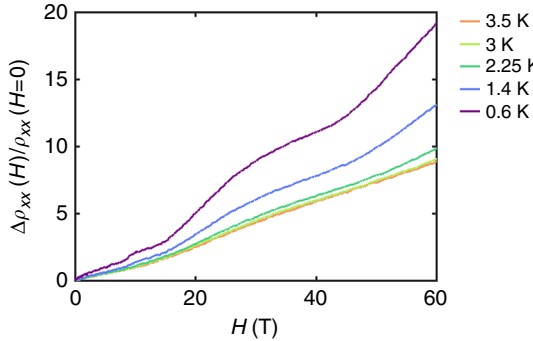

**Fig. 2** High field Shubnikov–de Haas oscillations. High field isothermal magnetoresistance curves of a 1.2-nm-thick LTO film on STO at low temperatures

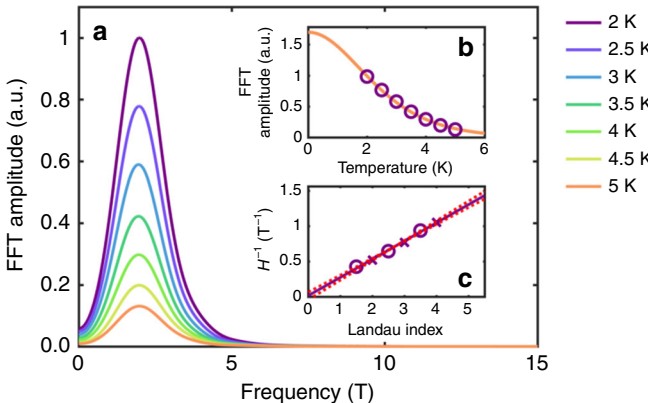

**Fig. 3** Analysis of low field oscillations. **a** The Fourier transform of the oscillations after a polynomial background was subtracted. The inset shows the temperature dependence of the peak amplitude. The orange line in the inset **b** is the fit to the Lifshitz–Kosevich equation which was used to determine the effective mass. **c** Landau level fan diagram shows maxima in $\rho_{xx}$ as crosses and minima as open circles

To isolate the oscillations in the ultra-thin samples, we subtracted a cubic spline fit for both the low and high field data (see Supplementary Note 2).The Fourier transform of the low field oscillations as a function of $1/H$ is shown in Fig. 3. The frequency of these low field oscillations is clearly peaked at $3.7 \pm 0.7$ T. The center of the peak appears to shift slightly as the temperature increases. However, this is due to small differences in the fit and is well within the error of the measurement and analysis. According to the Onsager relation, the cross-sectional area of this orbit is $3.9 \pm 0.8 \times 10^{-4}$ Å$^2$ or $0.015 \pm 0.003\%$ of the first Brillouin zone. The low oscillation frequency means that the last quantum oscillation occurs at 3.7 T: beyond this field the system is in the "quantum limit" where all quasiparticles are in the last Landau level. The temperature dependence of the amplitude is shown in the inset of Fig. 3 and fitted to the Lifshitz–Kosevich equation[25],

$$\frac{\sigma_{xx}(H) - \sigma_{xx}(H = 0)}{\sigma_{xx}(H = 0)} \propto \frac{X}{\sinh(X)}, \quad (1)$$

where $\sigma_{xx}(H)$ is the longitudinal conductivity at a magnetic field $H$, $X = (2\pi^2 m^* k_B T)/(\hbar e H)$, $m^*$ is the effective mass, and $T$ is the temperature (see Supplementary Note 3). Using the average inverse magnetic field at which the oscillations occur as 1.33 T$^{-1}$, we can deduce from the Lifshitz–Kosevich equation an effective mass for the low field oscillations of $0.12 \pm 0.01 m_e$, where $m_e$ is the free electron mass. For the high field oscillations, too few oscillations are observed to take the Fourier transform so the frequency was calculated based on the positions of the maxima, and the effective mass was calculated based on the amplitude at 32 T. This analysis gave a frequency of 39 T and an effective mass $1.2 m_e$, which shows that these two sets of oscillations probe two distinct bands.

Another key feature is the amplitude of the oscillations. The amplitude of the low field oscillations increases initially as the field increases and disappears when the quantum limit is reached. However, the oscillation amplitude in the magnetoresistance appears to decrease at the highest field values for these low field oscillations. This effect is removed when converting to conductivity and removing the background. We plot magnetoconductivity and divide by the background magnetoconductance in Supplementary Note 3 to show that indeed the oscillation amplitude increases with increasing field as expected for SdH oscillations.

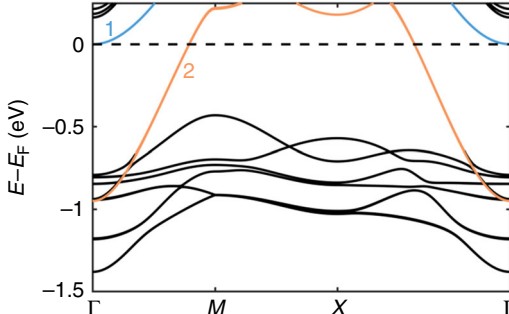

**Fig. 4** LTO/STO band structure. Band structure calculated with DFT + U for three monolayers of LTO on STO (001). The blue and orange bands (labeled 1 and 2, respectively) are close enough to the Fermi level to contribute to the electrical transport. The black bands are too far away to contribute any carriers

**DFT calculations and tight-binding model.** Complex oxide heterostructures often have complex band structures due to the contribution of multiple orbitals. So when analyzing SdH oscillations in these systems, it is necessary to take into account contributions from all of these orbitals. In our system, these orbitals include the Ti $d_{xy}$, $d_{xz}$, and $d_{yz}$ orbitals.

There have been a number of theoretical modeling studies of the LTO/STO interface. Some of these studies have been performed in the context of superlattices where the interfaces cannot be considered to be isolated interfaces[2,26] while others consider a single layer of LTO embedded in STO[27,28]. However these studies do not necessarily shed light on the electronic structure for these ultra-thin LTO films which show quantum oscillations. Therefore, we have theoretically modeled the electronic structure of 2 and 3 unit cells of LTO on STO (001) using generalized gradient approximation (GGA) + U calculations.

Density-functional theory (DFT) calculations shed insight into the bands that are relevant to conduction. Figure 4 shows the calculated band structure. There are clearly only two bands crossing the Fermi level that contribute to the electrical transport (bands 1 and 2 shown in blue and orange, respectively, in Fig. 4). Because the frequencies of the observed oscillations are very small, they cannot be attributed to the Fermi pocket created by band 2. DFT calculations indicate that band 2 comes from the interfacial $d_{xy}$ orbitals, while band 1 comes from the $d_{xz}$ and $d_{yz}$ orbitals in the STO (see Supplementary Note 4). Interestingly, band 1 is actually a hybridization of $d_{xz}$ and $d_{yz}$ orbitals rather than purely one or the other as is usually expected in STO. While $d_{xz}$ and $d_{yz}$ bands are typically highly anisotropic, this hybridization causes the band to be essentially isotropic near the $\Gamma$ point. Hybridization of the bands may be caused by distortions of the nearly cubic STO lattice in the presence of electric field from the polar discontinuity[29], and it has been shown to be enhanced with the addition of spin–orbit coupling[30,31]. In any case, both bands crossing the Fermi level appear to be isotropic.

The effective masses measured from the quantum oscillations must come from the bands which cross the Fermi energy in Fig. 4. These measured effective masses are compared with those calculated from the second derivative of the bands along the $\Gamma$–$X$ direction near the $\Gamma$ point from the DFT calculations. The calculated masses are $0.70 m_e$ for the $d_{xy}$ and hybridized $d_{xz+yz}$ bands, $4.1 m_e$ for the pure $d_{xz}$ band and $13.6 m_e$ for the pure $d_{yz}$ band. The effective mass of the hybridized band is nearly the same as the $d_{xy}$ bands near the $\Gamma$ point, but it increases at higher wave vectors due to an anisotropy which is not apparent at low wave vectors. If we naively attribute the higher frequency oscillations to pure $d_{xz}$ or $d_{yz}$ bands because of their larger

effective mass, then the effective mass deduced from DFT would be the geometric mean of the mass along the Γ–X and Γ–Y directions (i.e., $m^* = \sqrt{m_x m_y}$), thus predicting an effective mass of $7.4m_e$. This value is considerably larger than the measured cyclotron effective mass (defined as the derivative of the Fermi surface area). So we cannot attribute either of the oscillations to pure $d_{xz}$ or $d_{yz}$ bands. Moreover, both of the measured cyclotron masses are different from the calculated $d_{xy}$ and hybridized bands. However, a Rashba effect would create an inner band with a smaller cyclotron mass and an outer band with a larger cyclotron mass than the unsplit band. So the observed effective mass values are possibly consistent with quantum oscillations arising from a Rashba-type split $d_{xy}$ band or $d_{xz+yz}$ hybridized band.

The band structure in Fig. 4 shows that the $d_{xy}$ band (band 2) crosses the Fermi energy at approximately $X/2$, which is much larger than what is measured based on the frequencies of either of the oscillations. Therefore, the $d_{xy}$ band cannot give rise to the Fermi pockets measured by the quantum oscillations. On the other hand, the hybridized $d_{xz+yz}$ band (band 1) near the Fermi energy is near the Γ point so it can give rise to such a small Fermi pocket. Because two sets of oscillations are observed with only one possible band in Fig. 4 that can produce a Fermi pocket small enough to account for the frequencies, the simplest and most likely explanation is a spin splitting of this hybridized $d_{xz+yz}$ band.

In order to determine the details of the band structure near the Γ point with higher resolution, we have performed tight-binding calculations. Tight-binding calculations have shown that the $d_{xy}$, $d_{xz}$, and $d_{yz}$ orbitals all give rise to a linear Rashba-type splitting[29,32,33], so the above frequencies and effective masses can be used to estimate the Rashba-type coupling constant responsible for the splitting of the hybridized $d_{xz+yz}$ band. In order to properly model the properties of the hybridized $d_{xz+yz}$ band, we extended the tight-binding model described by Rödel et al.[34] to include a linear Rashba-type splitting (see Supplementary Note 5). Using this model, the energies of the resulting bands is given by $E(k) = \varepsilon_{xz+yz}(k) \pm \alpha|k|$, where $\alpha$ is the Rashba-type coupling constant and $\varepsilon_{xz+yz}(k)$ is the wave vector dependence of the energy without spin–orbit coupling. The hopping integrals of the tight-binding model were fit to the DFT band to determine $\varepsilon_{xz+yz}(k)$. Then, the Rashba-type coupling constant was adjusted to fit the effective masses and Fermi surface areas measured by the quantum oscillations. The measured effective masses of $0.12m_e$ and $1.2m_e$ are the cyclotron masses generally given by $m^*_\pm = \frac{\hbar^2}{2\pi} \frac{\partial A_\pm}{\partial E_F}$, where $E_F$ is the Fermi energy and $A_\pm$ are the areas of the inner and outer Fermi pockets. This form for the effective mass clearly shows why the outer surface has a larger mass because $\frac{\partial A}{\partial E_F}$ is larger for the outer surface. Based on this model, we can estimate a Rashba-type coupling constant of approximately $1.8 \times 10^{-11}$ eVm. A similar but larger constant of $5 \times 10^{-11}$ eVm has been found at the surface of cleaved STO[35]. For reference, the Rashba-type coupling constant is $3.8 \times 10^{-10}$ eVm in bulk BiTeI[36], $3.4 \times 10^{-12}$ eVm in LAO/STO[37], $6 \times 10^{-13}$ eVm in GaN[38], and $5 \times 10^{-15}$ eVm in SiGe/Si/SiGe quantum wells[39]. Additionally, this tight-binding model gives an effective mass of $0.66m_e$ for the unsplit band at the Γ point which matches that of the DFT calculated hybridized band as expected.

**Carriers from different bands**. A sheet carrier concentration of $1.8 \pm 0.3 \times 10^{11}$ cm$^{-2}$ was deduced from the low-frequency oscillations and a sheet carrier concentration of $1.7 \pm 0.3 \times 10^{12}$ cm$^{-2}$ from the high-frequency oscillations. Due to the nonlinear field dependence of the Hall resistance at these low temperatures, as shown in Fig. 1, a single carrier concentration value cannot be deduced from the Hall effect. This is in contrast to thicker films

where the Hall effect is linear. The Hall data in the ultra-thin films can be fit to a two-carrier model using the zero field resistivity as an additional constraint[40]. Such a two-carrier model has previously been applied to electrically gated, thick (>10 unit cells) LTO films on STO[21] and ungated δ-doped STO/LTO/STO superlattices[40]. In our case, we find a linear Hall effect in thick LTO films and a nonlinear Hall effect for the thinnest of LTO films and only at low temperatures. Our analysis of ultra-thin LTO films at low temperatures in terms of two-carrier types gives estimates of two sheet carrier concentrations of $1.9 \pm 0.2 \times 10^{14}$ cm$^{-2}$ and $2.0 \pm 0.2 \times 10^{12}$ cm$^{-2}$ with mobilities of $46 \pm 1$ cm$^2$/Vs and $(4.0 \pm 0.1)10^3$ cm$^2$/Vs, respectively (see Supplementary Note 6).

Given that neither of the carrier concentrations derived from the SdH oscillations frequencies match the high concentration carriers from the Hall effect fit, additional carrier types and hence a three band model may be appropriate. However, such a problem is not constrained enough to produce accurate or meaningful values. A three band model is easily simplified to a two band model if we assume that two of the bands have the same mobility. The model then becomes a two band model with one of the carriers having the concentration of the sum of the two similar mobility bands. In our case, the sum of the sheet concentrations deduced from the oscillations is equal to the low carrier concentration associated with high-mobility carriers deduced from the Hall effect, thus indicating that the oscillations account for all of the high-mobility/low-concentration carriers.

Comparing to the band structure, it appears that that these two-carrier concentrations are associated with bands 1 and 2 shown in Fig. 4 based on the Fermi wave vectors of the two bands; the low-concentration carriers come from $d_{xz+yz}$ hybridized orbitals of the Ti in the STO while the high concentration carriers come from the $d_{xy}$ orbitals of the Ti at the interface (see Supplementary Note 4 for the layer resolved density of states). The high concentration carriers deduced from the Hall effect appear to have too low of a mobility to give rise to oscillations even up to 60 T. Others have previously suggested that high-mobility carriers are associated with the filling of $d_{xz}$ and $d_{yz}$ bands[41] and our combined experimental and DFT results are consistent with these previous findings.

Together our experimental and theoretical results suggest that the different carriers come from $d_{xy}$ and $d_{xz+yz}$ hybridized bands associated with different parts of the heterostructure. There are three carrier types in total from two different bands which contribute to conduction. One band from the $d_{xy}$ orbitals on the interfacial Ti contributes the low mobility/high concentration carriers observed in the Hall effect (band 2 in Fig. 4). Note that we are unable to measure the quantum oscillations from this band so we are unable to determine if this band is Rashba-type split. The other two types of carriers come from the $d_{xz+yz}$ hybridized band in the STO (band 1 in Fig. 4) that is Rashba-type split into an inner and outer Fermi surface. This band contributes the high-mobility/low-concentration carriers observed in the Hall effect and both of the Fermi pockets observed in the quantum oscillations.

**Berry phase in LaTiO₃/SrTiO₃**. SdH oscillations can also be used to experimentally detect a Berry phase[25,42]. According to the Lifshitz–Onsager quantization rule, the Berry phase can be found by extrapolating a plot of $1/H$ vs. the Landau index, known as a Landau level fan diagram, to $1/H = 0$. The Lifshitz–Onsager quantization rule is as follows:

$$A \frac{\hbar}{eH} = 2\pi \left( n + \frac{1}{2} - \frac{\phi}{2\pi} \right), \qquad (2)$$

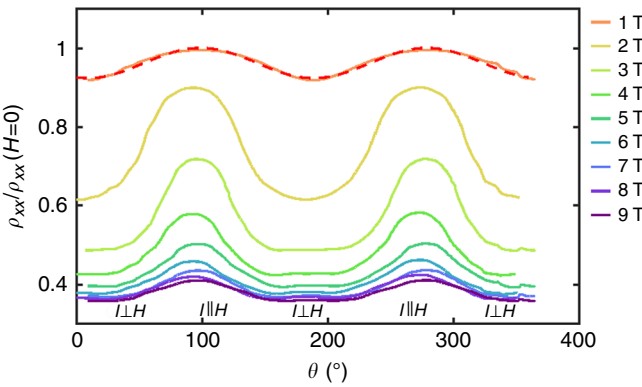

**Fig. 5** In-plane anisotropic magnetoresistance. Angular dependence of the in-plane magnetoresistance showing the AMR in various magnetic fields. The red dashed line shows a fit to $\cos^2\theta$ for the lowest field measurement

where $\phi$ is the Berry phase of the orbit in $k$-space, $H$ is the strength of an applied external magnetic field, $A$ is the extremal cross-sectional area of the Fermi surface perpendicular to the magnetic field, and $n$ is the Landau level index.

Figure 3c shows such a fan diagram for the low field oscillations. Clearly, the $x$-axis intercept of the fit line is $\sim 0$, which means that the Berry phase is $\sim\pi$. This result is reproducibly observed in multiple samples. A detailed error analysis[25] shows that the error is only 0.4 rad, which is too small to account for the observed phase. This error is surprisingly low given that only five extrema are used, but this is possible because of the small frequency of the oscillations. The fan diagram also clearly shows that the oscillations follow a $1/H$ dependence as expected from SdH oscillations, a dependence not immediately obvious from looking only at the raw data. Unfortunately, too few oscillations are observed in the high field measurements to perform such an analysis for the high-frequency oscillations. However, a direct fit to the Lifshitz–Kosevich equation gives a Berry phase of 3 rad.

In general, a nonzero Berry phase is the result of a band crossing. For the case of a perfectly linear Dirac point, the Berry phase is $\pi$ but is reduced and vanishes as a gap is introduced[43]. For systems that are described by a Rashba Hamiltonian, the point where the spin-split energy bands cross should correspond to a Dirac point. As a result, a Berry phase of $\pi$ may be realized for charge carriers that have an orbit around the Dirac point[44–49]. Our extracted value of $\sim\pi$ therefore indicates a Rashba-type splitting induced band crossing with a small, if any, gap. Well separated oscillations and a nonzero Berry phase have been observed in the bulk Rashba semiconductor BiTeI[42]. However in typical Rashba systems, the two frequencies are similar so the result is a beating pattern[50,51] which prevents an accurate extraction of the Berry phase. To date, there has been no report of such Berry phase in complex oxide interface systems.

**Weak anti-localization and AMR.** Moreover, the magnetoresistance is positive and has a shape reminiscent of weak anti-localization. The global shape of the magnetoresistance is fit quite well by a weak anti-localization model[52–54] (see Supplementary Note 7). This is consistent with a large Rashba effect at low temperatures. A value of $3 \times 10^{-3}$ ps was obtained for the spin–orbit relaxation time, which is significantly smaller than the scattering time of 1.5 ps obtained from the mobility of the carriers in our LTO/STO samples. This is in contrast to the LAO/STO system where typical values of the spin–orbit relaxation time and scattering times are comparable at 1 ps[5]. A Rashba-type coupling constant of $2.0 \times 10^{-11}$ eVm is deduced from the spin–orbit

relaxation time obtained from this model and is in excellent agreement with the value obtained from the two sets of oscillations.

Additionally, a very strong positive in-plane AMR was observed in the ultra-thin LTO samples. Figure 5 shows the angular dependence of the in-plane magnetoresistance for various magnetic field strengths at 3 K. As the magnetic field is increased, the anisotropy becomes stronger before reaching a maximum around 2.5 T and then decreasing and eventually saturating at high fields (see Supplementary Note 8). If we define the strength of the AMR as

$$\text{AMR} = \frac{\rho_{xx}(90°) - \rho_{xx}(0°)}{(\rho_{xx}(90°) + \rho_{xx}(0°))/2}, \quad (3)$$

then the maximum strength of the AMR is just over 40% and the saturating value is about 18%. Note that in this setup, 90° corresponds to the situation where the current is parallel to the field and 0° where the current is perpendicular to the field.

Large positive in-plane AMR has been thought to be consistent with a Rashba system[55,56]. It has been suggested that this Rashba effect is enhanced by the polar interface as is the case for the LTO/STO interface and is much smaller in nonpolar heterostructures[55]. In the LAO/STO system, large positive in-plane AMR has been found in some[57,58], but not all, samples. Moreover, it is predicted that the angular dependence for this AMR in a Rashba system is $\cos^2\theta$, where $\theta$ is the angle between the current and the magnetization[59]. However, AMR studies of LAO/STO have shown the angular dependence to be much more complicated and may be dominated by other magnetic effects.

## Discussion

Since quantum oscillations have never been reported in LTO thin films before, it is important to rule out any other sources for the oscillations. Previously, SdH oscillations have been seen in similar systems such as LAO/STO[60–63], $Al_2O_3$/STO[64], and GdTiO$_3$/STO[65] heterostructures, oxygen deficient STO[66], and La-doped[67] and Nb-doped[66,68,69] STO. However, there are some striking differences between these systems and the oscillations shown here. The clearest difference is that in the other systems the oscillations do not appear to reach the quantum limit with fields up to 9 T. Such a low frequency suggests that conduction at the LTO/STO interface is fundamentally different in nature from that of LAO/STO. Another possibility is that these oscillations come from a size effect such as Sondheimer oscillations because they only appear in the thinnest samples. However, oscillations due to size effects would not follow the Lifshitz-Kosevitch temperature dependence and are typically periodic in $H$ rather than in $1/H$ like the oscillations observed here. Furthermore, neither the oscillations nor the AMR show any hysteresis, and SQUID magnetometry measurements show no signs of ferromagnetism or hysteresis. Additionally, XMCD measurements at 10 K show no signs of magnetism or common magnetic impurities such as Fe or Co. Therefore, the nonlinear Hall effect, oscillations, and AMR are not likely caused by long-range magnetic ordering or magnetic impurities.

Now that it has been established that the oscillations are intrinsic to the heterostructure and not caused by magnetic ordering, it is important to consider the origin of such a small pocket in the Fermi surface. There are several possible scenarios which can explain multiple quantum oscillations in these complex oxide heterostructures. First the oscillation could come from an $d_{xy}$ band while the other comes from a pure $d_{yz}$ or $d_{xz}$ band. However, as explained previously, the effective mass of the low-frequency oscillations is much smaller than what we would expect for a pure $d_{yz}$ or $d_{xz}$ band and this situation does not explain the nonzero Berry phase. Second, the oscillations could come from

$d_{xy}$ bands from different LTO and STO layers. However, in this scenario the effective masses of the oscillations should be the same which is not what we observe, and again this does not explain the Berry phase. A third scenario to explain this data would be topological edge states. This would indeed explain the Berry phase, the AMR, and the weak anti-localization. However, we have found no theoretical justification to make such a claim. The fourth possible explanation is a giant Rashba effect which is the simplest explanation consistent with all the data presented here. As mentioned previously, the LTO/STO interface is polar which means that there is a large potential gradient near the interface which gives rise to an electric field[2]. Therefore, near the interface there can be a large Rashba-type interaction. Additionally, structural effects such as buckling at the interface can greatly enhance such a Rashba-type effect[70]. Our DFT results show evidence for a buckling near the interface in ultra-thin films. So it is not surprising that we observe such large Rashba-type effects. Because the two oscillations are well separated from each other, the strength of the Rashba effect must be anomalously large. The large Rashba-type splitting is consistent with the observed nonzero Berry phase for both the high- and low-frequency oscillations. Additionally, because the inner Fermi surface is so small, the Fermi energy must be close to the Dirac point induced by the splitting.

The idea of a giant Rashba-type coupling at the LTO/STO interface is also supported by the observed AMR. It has been predicted that the Rashba effect causes an AMR that increases in strength with increasing Rashba-type coupling[55]. The strength should also increase with increasing magnetic field until reaching a maximum and then decreasing as seen in our measurements[56]. Also, at low magnetic fields, the AMR follows a $\cos^2\theta$ angular dependence as expected for a Rashba system. Wang et al.[56] found with numerical calculations that a Rashba 2D electron system with a large Rashba-type coupling constant of $2\times10^{-11}$ eVm has a maximum AMR strength of nearly 22% but is strongly dependent on carrier concentration. The maximum strength observed in the LTO/STO system is over 40%, indicating that the Rashba-type coupling constant estimated from the SdH oscillations and the weak anti-localization fit are consistent with the observed AMR.

In summary, we have found evidence for a large Rashba-type coupling in films of LTO on STO. The low field oscillations and its associated Berry phase can be explained by a giant Rashba-type coupling which creates a small pocket in the Fermi surface. The two sets of quantum oscillations correspond to inner and outer Fermi surfaces indicative of a Rashba-type spin splitting. This is consistent with the observed AMR as both the size and magnetic field dependence have been theoretically predicted for a system with a very strong Rashba effect. Additionally, the field dependence of the magnetoresistance follows a weak anti-localization model. Such a large Rashba-type coupling provides an excellent avenue to explore further possibilities in the field of spintronics using LTO films.

## Methods

**Sample growth.** LTO films were grown on STO single crystal substrates by pulsed laser deposition. A KrF laser ($\lambda$=248 nm) operated at a repetition rate of 1 Hz was incident upon a $La_2Ti_2O_7$ target with a fluence of 1.4 J/cm$^2$. The substrate was heated to 625 °C in $10^{-6}$ Torr. Previous studies have shown that the $LaTiO_3$ phase can be stabilized on a substrate from a $La_2Ti_2O_7$ target under the appropriate conditions[71]. Commercially available (001)-oriented STO substrates from Crystec were used. Prior to the deposition, the substrates were treated by chemical etching to achieve a $TiO_2$-terminated surface[72], and then annealed at 1000 °C in atmosphere. The films were between 1.2 and 60 nm thick. Above ~9 nm, the thickness was determined by x-ray reflectivity (XRR). Below ~9 nm, the films were too thin to measure with XRR so the thickness was estimated by extrapolating from the growth rate of the thicker films. Substrates were subject to the growth conditions without film deposition to verify that they were not reduced in the process and did not contribute to the electronic conduction.

**Electrical measurements.** The electrical transport measurements were taken using a Hall bar geometry of dimensions 5 mm × 1 mm with ultrasonically wire-bonded aluminum wires as electrodes. The temperature and magnetic field dependence were measured in a Quantum Design Dynacool system from 2 to 300 K and up to an applied magnetic field of 9 T. Additionally, magnetoresistance measurements were performed up to 60 T on a 1.2-nm-thick sample at the National High Magnetic Field Laboratory of Los Alamos National Laboratory. Because the magnetoresistance is an even function of applied field, the resistance was averaged over positive and negative fields to remove any residual Hall effect in the longitudinal magnetoresistance measurements. Similarly, the difference between the Hall resistance at positive and negative fields was taken to remove any residual longitudinal magnetoresistance in the Hall effect measurements.

**DFT calculations.** DFT calculations were performed using the full-potential linearized augmented plane wave code WIEN2k[73]. The exchange-correlation potential was calculated with the GGA[74] and the fully localized and rotationally invariant "+U" correction[75,76]. The $U_{eff}$ ($=U-J$) parameter has been set to 5 and 7 eV, respectively, for the $3d$ electrons of Ti atoms, and the $4f$ electrons of La atoms. A $c$ (2×2) lateral unit cell was used containing two non-equivalent Ti sites per atomic layer was used in order to take into account of possible structural distortions, with a lateral lattice parameter set to the GGA equilibrium lattice constant of STO (3.92 Å, slightly larger than the experimental value 3.905 Å). The atomic positions were fully relaxed. In order to avoid the emergence of a spurious electric field due to the periodic boundary conditions, we have chosen a LTO(3MLs)/STO(1.5MLs)/LTO (3MLs)(001) symmetric slab with LTO layers on both sides of the STO substrate and a vacuum region between the slab and its periodic images of 20 Å. Although at each interface inversion symmetry is broken, the symmetry of the overall system and possible interference between the two interfaces does not allow us to properly model Rashba-type splitting. This aspect of the DFT calculations goes beyond the present study.

**Data availability.** All relevant data are available from the corresponding author upon reasonable request.

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

## Acknowledgements

We thank M.T. Gray for assistance with depositions as well as M. Gabay and X. Qi for insightful discussions. This work was funded by the Vannevar Bush Faculty Fellowship of the Department of Defense under Contract No. N00014-15-1-0045. M.J.V. acknowledges a National Science Foundation Graduate Fellowship. Work at the National High Field Magnetic Laboratory was funded by the state of Florida, the Department of Energy, and the NSF Cooperative Agreement No. DMR-1157490. This work was granted access to the HPC resources of the CALMIP supercomputing center under the allocation p1229. R.A. and R.P. acknowledge funding by the German Science Foundation within SFB/TRR 80 (projects C03 and G03) and computational time at the Leibniz Rechenzentrum Garching, project pr87ro.

## Author contributions

M.J.V. designed and performed experiments, analyzed data, and wrote the paper; B.J.R. provided support for the high magnetic field experiments; R.A. and R.P. performed and analyzed the DFT calculations; Y.S. designed experiments and wrote the paper.

## Additional information

**Competing interests:** The authors declare no competing interests.

