## [Peer Review File · Nature Communications]

Reviewers' comments:

Reviewer #1 (Remarks to the Author):

Veit et al. present magnetoresistance measurements including quantum oscillations for a LTO/STO interface and conclude a large Rashba coupling coefficient. Large Rashba effects have been reported before in other STO interfaces, see Refs. [5,6].

The weak spot of the paper is the rather indirect proof which heavily relies on the interpretation or modeling, and this interpretation is most certainly wrong: The authors use two copies of a simple one-band isotropic, free electron description. This might be a proper description for semiconductors but is not adequate for transition metal oxide heterostructures. First of all with the electronic reconstruction we can expect metallicity in both LTO and STO, see e.g. J. Phys.: Condens. Matter 22, 043001 (2010). As there is not only a single layer it is at least unclear whether not also xz and yz orbitals contribute besides the xy orbital - and they have largely different Fermi surfaces. At least for other STO interfaces the biggest spin splitting occurs when the orbitals cross, see Phys. Rev. B 87, 161102(R) (2013); 88, 041302(R) (2013). Because of the quantization in 3 layers, we have 3 different xy orbitals which are shifted with respect to each other and can be expected to have largely different Fermi surfaces. That is we need to expect many more Fermi surfaces, that are for other reasons than the spin orbit coupling largely different in size. A proper modeling is needed instead of borrowing equations which have been developed for a completely different situation.

I cannot recommend the publication of the manuscript in Nature Communications without properly taking into account even the most basic features of the electronic structure. This way the conclusions such as the strength of the spin orbit coupling are unavoidably wrong.

Reviewer #2 (Remarks to the Author):

This manuscript presents the first time observation of quantum oscillations in the longitudinal magneto-conductance of LaTiO₃/SrTiO₃ interface which the authors have suggested as a manifestation of Shubnikov de Haas Oscillations of the two dimensional electron gas formed at the interface. These SdH oscillations, combined with the ubiquitous positive magnetoresistance in LAO/STO and LTO/STO which has been interpreted by many groups in the framework of

weak antilocalization, have led the authors to conclude a dominant Rashba spin – orbit coupling in the 2DEG. While the observation of SdH in LTO/STO is new and surely worth reporting, the authors ignorance of the existing state of the art of magneto-transport in LTO/STO systems, which is described extensively in the following papers, comes as a surprise. Some of the papers are;

Phys. Rev. Lett. 108, 247004(2012)

Phys. Rev. B 94, 115165 (2016)

Phys. Rev. B 90, 081107(R) (2014)

For example, the observation of non-linearity of high field Hall data on field, its reversible nature and its analysis in the framework of a two-band model have been done extensively in PRL 108, 247004 . The results presented in this manuscript are a confirmation of that study. Further, the analysis of magneto conductivity in the framework of weak antilocalization has been done for a variety of LTO/STO interfaces in PR 94, 115165. It is surely worthwhile to compare the spin – orbit scattering time/energy given in this manuscript with the earlier reports on LTO/STO and LAO/STO.

Further, the authors have shown a distinct Berry Phase contribution to magneto-transport. It is important to see how this non-zero Berry Phase contributes to the Hall Effect and its non-linearity on field. In a Berry Phase scenario, would it be necessary to invoke a two-band picture for the non-linear Hall? It is also desirable to discuss the two-band model in the framework of the existing knowledge of the electronic structure of the interface, particularly when the authors claim that the Rashba coupling creates a small pocket in the Fermi surface.

Reviewer #3 (Remarks to the Author):

The manuscript "Quantum oscillations resulting from giant Rashba spin splitting in LaTiO₃/SrTiO₃ heterostructure", by M. J. Veit, et al., seems to contain some interesting results that may be worth publishing.

However, there are several issues that do not allow for a definite statement about publication to be made.

First of all, the authors do not specify whether their sample is superconducting or not (at yet lower temperatures than those explored), and what is the critical temperature of their sample. This information is crucial to compare their sample to known measured samples.

The second aspect is gating. One striking property of these oxide heterostructure is gate-tunability (e.g., the Rashba spin orbit coupling increases with increasing gating), so I was surprised that the authors did not try to follow the evolution of their measurements with gating.

Is there any fundamental reason why this important perspective cannot be given?

The third aspect is comparison to previous results. When reading the text, the reader has the impression that LaTiO₃/SrTiO₃ is as yet poorly studied, which is not really the case. When trying to compare the present results with results published long ago (e.g. PRL 108, 247004) I was quite confused, as the mobility of the high-mobility carriers is reported here to be ten times larger, whereas the mobility of the low-mobility carriers seems definitely smaller (by a factor of two or so). Even more surprisingly, the carrier density of the low-mobility carriers seems to be more than 20 times larger here, whereas the reported high-mobility carrier concentration is far smaller (almost two orders of magnitude!). Of course, the analysis of PRL 108, 247004 is much more complete within this respect, because the evolution of these numbers with gating is reported, and an overall coherent scenario is provided, which is not the case here. Therefore, addressing the origin and meaning of such discrepancies is mandatory.

The results of PRL 108, 247004, as well as many other experimental and theoretical results (see, e.g., PRL 116, 026804), suggest that several sub-bands are formed within the potential well confining the electron gas at the interface. In particular, it has been suggested that the appearance of high-mobility carriers might be related to the filling of xz and yz sub-bands. However, no reference whatsoever is made to such a multi sub-band structure, nor to its consequences on the interpretation of the present results. For instance, can the interpretation of a surprisingly larger Rashba spin-orbit coupling be biased by neglecting this multi sub-band structure?

There are also some minor points.

- a. I noticed that the first author's name is misspelt in Ref. [23], and the list of authors of Ref. [38] seems incomplete.
- b. In the supplemental material, the first equation of Sec. II contains an \exp and a \cos symbol that are not properly displayed (they should be typed as $\backslash\exp$ and $\backslash\cos$ in the tex file).
- c. The caption of Fig. S3 contains a typo, "it" for "fit".
- d. A citation to Ref. [38] is missing in Sec. IV of the supplemental material.
- e. When discussing the fit in Fig. S6, the value of the Rashba spin-orbit coupling is given with a repetition of the units of measurements.

I cannot exclude that a properly revised version of the manuscript may meet the criteria for publication in Nature Communications, but I suggest that this piece of work cannot be accepted for publication in the present form.

Reviewer #4 (Remarks to the Author):

The authors report on three types of magnetotransport measurements: The Shubnikov-de Haas effect, weak anti-localization correction to the conductivity and anisotropic magnetoresistance in very thin films of LaTiO₃ deposited on SrTiO₃. The authors suggest that all these measurements are consistent with a very large Rashba spin-orbit coupling, the largest known in oxide systems, of the order of 1.6 10⁻¹¹ eV-m.

Engineering systems with large spin-orbit coupling is an important task for future spintronics applications. The data presented in this thorough study are very interesting and the conclusions are intriguing.

I would recommend publication in Nature Communications after the authors have answered the following questions:

Perhaps the most intriguing finding is the π Berry's phase obtained from the low field oscillations. Setting the error bar on this phase is tricky. In fact, the phase is a function of the frequency itself and of the magnetic field due to the Zeeman effect. The latter seems to be less important in the presented data since the fields are relatively low and the Rashba splitting is presumably very large. I would suggest to try and use various frequencies around the maximum in FFT (within the error of the frequency) and see how much the extrapolation in the inset of figure 3 depends on the frequency or on the inferred area of the Fermi-surface

The authors claim that the SdH signal vanishes for films thicker than 2nm. Is this an abrupt effect? What happens for 2nm versus 2.4nm?

Can the authors demonstrate that the SdH effect is 2D in nature? E.g. by tilting the field and observing a $1/\cos 2\theta$ dependence.

In line 160 I believe that it should be 1013.

Do I understand correctly that from the Hall and SdH analysis the authors obtain three bands: The first observed in low field SdH with 10¹¹, another one with 10¹² both detected by the SdH as well and a third band that accommodates the majority of the carriers, which is not seen by SdH but dominates the Hall. How come the 10¹² band is not observed in the Hall and the 10¹¹ band is? The authors should clarify this point.

The authors write "large positive in-plane AMR has been found in some [41, 42], but not all, samples." Is it true that large and positive AMR is observed in polar oxide interfaces while it is absent in non-polar ones as suggested in ref. 42? This is in fact consistent with the current paper.

Finally, how strong the obtained Rashba coupling comparing to the Ti atomic spin orbit coupling?

Responses to Referees

We describe how we have addressed all of concerns of all four referees and articulate the revisions in detail below:

REFEREE 1

Referee 1 has pointed out that “*the authors use two copies of a simple one-band isotropic, free electron description. This might be a proper description for semiconductors but is not adequate for transition metal oxide heterostructures.*” He/she then goes on to explain that in the case of LTO/STO, one needs to take into consideration multiple orbitals from xy as well as of xz and yz character that cross the Fermi level and that therefore our “*interpretation is most certainly wrong.*”

We realized from the referee’s comments that explicit theoretical justification of our simplified assumptions should have been included in our manuscript. We certainly agree with the referee that in transition metal oxide heterostructures such as $\text{LaTiO}_3/\text{SrTiO}_3$ we must consider multiple bands crossing the Fermi level. We agree with the referee that the metallic conduction can be found both in the LaTiO_3 and SrTiO_3 sides of the interface as indicated in J. Phys. Cond. Matt. 22 043001 (2010). Certainly, other perovskite oxide interfaces that have been theoretically studied in the context of Rashba spin-orbit coupling have taken the quantization of the xy orbitals as well as the xz and yz orbitals into account. Therefore, we agree with the referee that multiple bands of different character (i.e., xy , xz and yz) must be taken into account when analyzing electronic transport measurements.

To our knowledge, there have been several theoretical modeling studies of the $\text{LaTiO}_3/\text{SrTiO}_3$ interface but fewer compared to the $\text{LaAlO}_3/\text{SrTiO}_3$ interface. Several of these studies have been performed in the context of $\text{LaTiO}_3/\text{SrTiO}_3$ superlattices where the interfaces cannot be considered isolated interfaces [PRL **94** 176805 (2005), PRL **99** 016802 (2007)] while others consider a single layer of LaTiO_3 embedded in SrTiO_3 [PRB **73** 195403 (2006), PRB **77** 245152 (2008)]. However, these studies do not necessarily shed light on the electronic structure for our system of 2-3 unit cells of LaTiO_3 film on SrTiO_3 .

Therefore, we have now theoretically modeled the electronic structure of 2 and 3 unit cells of LaTiO_3 on SrTiO_3 . We now include GGA+U calculations of the density of states specifying the xz , yz , and xy character of the Ti 3d bands at and near the Fermi level. These calculations show that there is essentially no contribution from the xz and yz bands to the density of states near the Fermi level. Only three parabolic xy bands that have nearly equal effective masses and are offset from each other contribute to the conduction. As explained in Phys. Rev. B **87**, 161102(R) (2013), bands with xy character show a linear Rashba splitting as in our model.

The GGA+U calculations were used to explore in more detail the effective masses of the different bands near the Γ point. The xy bands have an effective mass of $0.7 m_e$. The xz and yz bands have an anisotropic effective mass with a mass of $0.61 m_e$ for the xz band and $8.0 m_e$ for the yz band along the Γ -X direction. In Shubnikov-de Haas oscillations, the effective mass observed in quantum oscillations for such an elliptical band should be the geometric

mean (i.e. the square root of the product) of the two effective masses. Therefore, the measured mass for the xz and yz bands should be on the order of $2.2 m_e$. This is significantly higher than either of our measured masses so the oscillations cannot be explained in terms of the xz or yz bands, consistent with the conclusions from the GGA+U calculations of the density of states. Moreover, a closer look at the calculated effective masses of the xy bands (contributing to the conduction) reveal nearly equal effective masses. However, the two measured effective masses from our quantum oscillations are vastly different from each other and therefore we cannot be attributed to two different xy bands. In a Rashba splitting scenario, averaging the observed effective masses gives the effective mass of the band without the Rashba effect. The average of our two measured effective masses is $0.66 m_e$. This value matches the calculated value for the unsplit xy bands.

Therefore, the prediction from DFT calculations that there are only xy bands at the Fermi surface and the large difference in the two effective masses deduced from the oscillations justify our simplifying assumption of considering a single Rashba split band.

On page 3 paragraph 1 through 3, we now justify our simplifying assumption of considering only parabolic bands of xy character by including a new figure (Figure 4) of the density of states indicating the absence of xz and yz character at the Fermi level and a significant discussion of the GGA+U calculations in the context of effective masses.

REFEREE 2

1. Referee 2 states that our work presents the “first time observation of quantum oscillations in the longitudinal magneto-conductance of LaTiO₃/SrTiO₃ interface which the authors have suggested as a manifestation of Shubnikov de Haas Oscillations of the two-dimensional electron gas formed at the interface.” He/she then goes on to state that “while the observation of SdH in LTO/STO is new and surely worth reporting, the authors ignorance of the existing state of the art of magneto-transport in LTO/STO systems, which is described extensively in the following papers, comes as a surprise.”

We are aware of the numerous papers from the Lesueur/Budhani collaboration on the electronic transport of the LaTiO₃/SrTiO₃ system that the referee listed. We chose a representative paper published in Nature Communications from their group. However, in the interest of completeness, we now include the PRL article (PRL **108** 247004 (2012)) and the Scientific Reports article (Sci. Rep. **4** 6788 (2014)) whose gating study has some overlap with the earlier PRL 108 247004 (2012) that the referee cited.

Furthermore, the referee states that PRB 94 115165 (2016) and PRB 90 081107 (2014) also should have been cited. We note that both studies are focused on the delta-doping of the LaTiO₃/SrTiO₃ interface with LaCoO₃ and LaCrO₃ respectively. Since the insertion of these layers inevitably give rise to disorder as well as magnetic scattering centers, we did not compare our samples with delta-doped samples. However, both studies do include magnetoresistance data of control samples of 20 unit cells of LaTiO₃ films grown on SrTiO₃ with no delta doping. In LaTiO₃ films of this thickness, we have found in our previous studies (PRB **81** 161101 (2010) and PRB **86** 081401 (2012)) that the entire LaTiO₃ film is

metallic. These thicker films are metallic due to an electronic structure modification from epitaxial compressive strain and the conduction is dominated by bulk conduction through the LaTiO₃ film. In our ultra-thin LaTiO₃ films, there is an additional contribution to the conduction from the interface itself that is detectable in the electronic transport measurements. Only in these ultra-thin films do we observe quantum oscillations.

We realized from the referee's comments that we had not clearly articulated the different film thickness regimes and how the ultra-thin regime shows different properties from previously reported results in the introductory paragraphs. We have now rewritten the introductory paragraphs describing the different behavior in thinner and thicker films and cite all additional references with respect to thicker films.

2. Referee 2 states that “the observation of non-linearity of high field Hall data on field, its reversible nature and its analysis in the framework of a two-band model have been done extensively in PRL 108, 247004 (2012). The results presented in this manuscript are a confirmation of that study.”

We agree with the referee and have now included the PRL reference as described above. We do not deny that there has been previous analysis of Hall data in the two-band model framework. We now make an appropriate citation to the PRL at this point in our manuscript. However, we note that in PRL 108, 247004 (2012) and other similar studies, a nonlinear Hall effect is observed only when the samples are gated. Here we observe a nonlinear Hall effect in films thinner than 2 nm *without any gating*. In our thicker than 2 nm films, we see a linear Hall effect which is indeed consistent with prior results. So, we see a thickness dependent nonlinearity which has not been seen before in LTO/STO heterostructures rather than a gate bias dependent nonlinearity. Additionally, the true novelty of our work is that we observe quantum oscillations in the Hall and longitudinal magnetoresistance data; to our knowledge, this is the first observation of quantum oscillations in the LaTiO₃/SrTiO₃ system. Furthermore, we observe a Berry's phase in the Shubnikov de Haas oscillations that has not been observed in other transition metal oxide heterostructures. As referee 4 comments, the observation of a Berry's phase is “perhaps the most intriguing finding.” *Therefore, we believe that our study is not a mere confirmation of the PRL study as the referee suggests.*

3. Referee 2 iterates that “the analysis of magneto conductivity in the framework of weak antilocalization has been done for a variety of LTO/STO interfaces in PR 94, 115165. It is surely worthwhile to compare the spin – orbit scattering time/energy given in this manuscript with the earlier reports on LTO/STO and LAO/STO.”

We thank the referee for pointing this out. We had carried out the calculations for spin-orbit scattering times from the expression for the weak antilocalization correction to the magnetoconductivity (a modified form of the Maekawa-Fukuyama formalism cited in reference 38 (Sci. Rep. 5 12751 (2015) as well as in PRB 94 115165 (2016)). We obtained 3×10^{-3} ps for the spin orbit scattering time which are significantly smaller than the typical spin-orbit scattering times of 1 ps obtained for LaAlO₃/SrTiO₃ (PRL 104, 126803 (2010)) and comparable to 10^{-3} ps found in PRB 94 115165 (2016) for LTO/STO. The difference between the previous report of the scattering time in LTO/STO and our result is likely

because they used an estimated effective mass of $3 m_e$ and we have used the actual effective mass from the quantum oscillations. Instead of comparing the spin-orbit scattering times, in our manuscript we had opted to compare the Rashba coefficients of our $\text{LaTiO}_3/\text{SrTiO}_3$ with those of $\text{LaAlO}_3/\text{SrTiO}_3$. However, for completeness, we now include our spin orbit scattering times and point out the significant difference compared to spin-orbit scattering times of $\text{LaAlO}_3/\text{SrTiO}_3$.

We now include a statement about the spin-orbit scattering times estimated from the weak anti-localization correction to the magnetoconductivity on page 5 paragraph 1.

4. Referee 2 asks whether “in a Berry Phase scenario, would it be necessary to invoke a two-band picture for the non-linear Hall? It is also desirable to discuss the two-band model in the framework of the existing knowledge of the electronic structure of the interface, particularly when the authors claim that the Rashba coupling creates a small pocket in the Fermi surface.”

Referee 2 questions whether a Berry phase can give a nonlinear Hall effect. To our knowledge, a Berry phase alone cannot give a nonlinear Hall effect. A well-studied example of this is graphene whose electrons do show a Berry phase, but the Hall effect is linear. Additionally, unlike in other studies where a two-band model is deduced solely from the non-linearity in the Hall effect measurements, we observe two Fermi pockets directly through the Shubnikov de Haas oscillations at low fields and high fields in addition to the nonlinear Hall effect data, and the frequency of the oscillations clearly shows that there are more carriers measured by the Hall effect than by the oscillations. Therefore it is clear that the nonlinear Hall effect cannot be entirely attributed to a single band. In order to discuss the band structure in more detail, we also now include GGA+U calculations which shows bands of only xy character at the Fermi level. There are several xy bands crossing the Fermi level each coming from different layers in the heterostructure. Therefore, we believe that the two carrier types are coming from different locations (i.e. one from xy bands in the STO and one from xy bands in the LTO). We have included a discussion of this in the updated manuscript on page 4 paragraph 2.

REFEREE 3

We thank referee 3 for noting that we have “interesting results” but that a few issues need to still be addressed. We list the issues below and address them one by one.

First of all, the authors do not specify whether their sample is superconducting or not (at yet lower temperatures than those explored), and what is the critical temperature of their sample. This information is crucial to compare their sample to known measured samples.

We would like to point out that we have found two previous studies from the same group (Nat. Commun. 1, 89 (2010) and PRL 108, 147004 (2012) with such low temperature transport data. In one of their studies, 10 and 15 unit cells of LaTiO_3 on SrTiO_3 single crystal substrates exhibited a superconducting transition temperature of 260 mK and 310 K respectively. In their second study, 10 unit cells of LaTiO_3 on SrTiO_3 single crystal substrates did not show a superconducting transition without any electrical gating. Given the fact that

oxygen deficient SrTiO₃ also exhibits a superconducting transition around these temperatures and that the samples were grown in an oxygen partial pressure of 1×10^{-4} Torr, it is difficult to separate out the contributions from oxygen deficient SrTiO₃ and the LaTiO₃/SrTiO₃ interface. In our case, we have found that in the one sample that we cooled down to 50mK in a dilution refrigerator, we did not observe a superconducting transition.

The second aspect is gating. One striking property of these oxide heterostructure is gate-tunability (e.g., the Rashba spin orbit coupling increases with increasing gating), so I was surprised that the authors did not try to follow the evolution of their measurements with gating. Is there any fundamental reason why this important perspective cannot be given?

We are in the process of developing gated samples. However, since we are limited in access to the high field magnetic field facility, gating measurements will not be possible for inclusion on the timescale of this manuscript. These experiments will be reported in a future report. We believe that we have sufficient experimental evidence, including two sets of quantum oscillations, Berry's phase, weak antilocalization correction to the magnetoconductivity and anisotropic magnetoresistance, to support a large Rashba spin-orbit coupling.

The third aspect is comparison to previous results. When reading the text, the reader has the impression that LaTiO₃/SrTiO₃ is as yet poorly studied, which is not really the case. When trying to compare the present results with results published long ago (e.g. PRL 108, 247004) I was quite confused, as the mobility of the high-mobility carriers is reported here to be ten times larger, whereas the mobility of the low-mobility carriers seems definitely smaller (by a factor of two or so). Even more surprisingly, the carrier density of the low-mobility carriers seems to be more than 20 times larger here, whereas the reported high-mobility carrier concentration is far smaller (almost two orders of magnitude!). Of course, the analysis of PRL 108, 247004 is much more complete within this respect, because the evolution of these numbers with gating is reported, and an overall coherent scenario is provided, which is not the case here. Therefore, addressing the origin and meaning of such discrepancies is mandatory.

We certainly did not mean to give the impression that no other magneto transport studies of LaTiO₃/SrTiO₃ have been performed. Budhani's group have developed LaTiO₃/SrTiO₃ and provided samples to Lesueur's group over the years and it is this collaborative effort that has provided most, if not all, of the previous magneto transport data on LaTiO₃/SrTiO₃ (besides our own studies of very thick LaTiO₃ films on SrTiO₃ substrates – PRB 81, 161101 (2010) and PRB 86, 081401 (2012)). We merely meant that the number of studies on LaTiO₃/SrTiO₃ has been significantly fewer than the large number of research groups and associated studies on the LaAlO₃/SrTiO₃ system.

PRL 108 247004 reports that the carrier concentration and mobility values of 10 unit cells of LaTiO₃ on SrTiO₃ is on the order of $8-10 \times 10^{13} \text{ cm}^{-2}$ and $25-30 \text{ cm}^2/\text{Vsec}$ with zero bias in the range of 4-80K. In this study, we have focused on 2-3 unit cells of LaTiO₃ on SrTiO₃ that exhibit carrier concentrations on the order of $1.9 \times 10^{14} \text{ cm}^{-2}$ and $1.9 \times 10^{12} \text{ cm}^{-2}$ with mobilities of 366 and 4400 cm^2/Vsec respectively at zero bias. It is therefore difficult to compare the exact mobilities and carrier concentrations among these papers as the referee suggests

because we see a nonlinear Hall effect without any bias in films thinner than 2 nm. In our own previous studies of thicker LaTiO_3 films (2.5-60nm thick) on SrTiO_3 (PRB 81, 161101 (2010) and PRB 86, 081401 (2012)), we have found volume carrier concentrations on the order of $1 \times 10^{22} \text{cm}^{-3}$ which corresponds to a sheet carrier concentration on the order of $4 \times 10^{15} \text{cm}^{-2}$ for a 4nm or 10 unit cell thick film. These carrier concentration values correspond to about 0.8e-/Ti ion which is consistent with metal carrier concentrations. As we have described in the introductory section of our paper, there appears to be an additional contribution from the interface that becomes dominant around a 2nm film thickness where the resistance no longer scales with LaTiO_3 thickness. This interface contribution appears to be more resistive and is consistent with a lower carrier concentration compared to our thicker films. Therefore, the carrier concentration and mobility values of our thin and thicker LaTiO_3 films on SrTiO_3 are self-consistent.

It is not exactly clear what are the differences in the samples from the different groups. However, the exact oxygen content can change mobility and carrier concentrations in LaTiO_3 . It is well known that oxygen content can be easily varied in bulk LaTiO_3 . For example, see Z. Phys. B 82, 211 (1991) where it is shown that the oxygen content can be varied continuously from the optimal stoichiometry of 3 to a stoichiometry of 3.5 for the $\text{La}_2\text{Ti}_2\text{O}_7$ phase. This changes the average number of 3d electrons per Ti ion which changes the carrier concentration and induces various metallic and semiconducting phases. It appears that Biscaras et al. in PRL 108 247004 use a stoichiometric LaTiO_3 target in a pressure of 10^{-4} Torr of oxygen. Ohtomo et al. in APL 80, 3922 (2002) have found that this elevated pressure actually creates the $\text{La}_2\text{Ti}_2\text{O}_7$ phase instead of the LaTiO_3 phase. Our group has found that growing with an oxygen rich $\text{La}_2\text{Ti}_2\text{O}_7$ target (which is more easy to stabilize than a LaTiO_3 target) in a lower oxygen pressure of $\sim 10^{-6}$ Torr minimizes the amount of the $\text{La}_2\text{Ti}_2\text{O}_7$ phase in the films. Therefore, there are likely differences in the oxygen content between our films and the films in PRL 108 247004, thus causing the differences in carrier concentration and mobility.

Referee 3 also points out the issue of multiple bands as did Referee 1. The results of PRL 108, 247004, as well as many other experimental and theoretical results (see, e.g., PRL 116, 026804), suggest that several sub-bands are formed within the potential well confining the electron gas at the interface. In particular, it has been suggested that the appearance of high-mobility carriers might be related to the filling of xz and yz sub-bands. However, no reference whatsoever is made to such a multi sub-band structure, nor to its consequences on the interpretation of the present results. For instance, can the interpretation of a surprisingly larger Rashba spin-orbit coupling be biased by neglecting this multi sub-band structure?

We thank the referee for pointing this out and have attempted to address the issue of multiple bands with additional theoretical calculations. As we point out in our response to referee 1, we find through our DFT calculations of 2 and 3 unit cells of LaTiO_3 on SrTiO_3 that the only bands that cross the Fermi level are xy bands, and the xz and yz bands are filled. Therefore, the bands that contribute to the conduction and oscillations are parabolic bands with equal effective masses that are offset in energy from each other. Because we see two very different effective masses from the oscillations, we cannot contribute the oscillations to different xy

bands. Also, neither of the masses are large enough to be from xz or yz bands. Therefore, we have added these DFT calculations to justify our simplifying assumption of a single Rashba split band.

In our thick samples, the sheet carrier concentration and sheet resistance scale with the film thickness and mobilities are comparable to the mobility of the low-density carriers (PRB **81** 161101 (2010) and PRB **86** 081401 (2012)). In ultra-thin films (<2 nm thick), the sheet resistance no longer scales with the thickness and we attribute additional conduction to the interface which is shorted out and therefore not observed in the thick samples. So, rather than attribute high mobility carriers to different types of sub-bands as the referee suggests, we attribute the high mobility carriers to those in the LTO, and the low mobility carriers to those at the interface where there is likely more impurities and impurity scattering. This is consistent with the DFT results which show bands from the LTO layers, the STO layers, and the interface layer all contributing to the conduction.

Additional points by referee 3 include:

a. *I noticed that the first author's name is misspelt in Ref. [23], and the list of authors of Ref. [38] seems incomplete.*

The spelling and the list of authors have been corrected in the references.

b. *In the supplemental material, the first equation of Sec. II contains an exp and a cos symbol that are not properly displayed (they should be typed as \exp and \cos in the tex file).*

The functions are now correctly displayed.

c. *The caption of Fig. S3 contains a typo, "it" for "fit".*

The caption has been corrected for Figure S3.

d. *A citation to Ref. [38] is missing in Sec. IV of the supplemental material.*

The citation is now included.

e. *When discussing the fit in Fig. S6, the value of the Rashba spin-orbit coupling is given with a repetition of the units of measurements.*

The typo has been corrected.

REFEREE 4

We thank the referee for his/her comments stating that “*data presented in this thorough study are very interesting and the conclusions are intriguing.*” We appreciate his/her positive response and recommendation for publication after several issues are addressed.

Referee 4 states that “perhaps the most intriguing finding is the π Berry’s phase obtained from the low field oscillations. Setting the error bar on this phase is tricky. In fact, the phase is a function of the frequency itself and of the magnetic field due to the Zeeman effect. The latter seems to be less important in the presented data since the fields are relatively low and the Rashba splitting is presumably very large. I would suggest to try and use various frequencies around the maximum in FFT (within the error of the frequency) and see how much the extrapolation in the inset of figure 3 depends on the frequency or on the inferred area of the Fermi-surface.”

We thank the referee for their suggestion on finding the error on the Berry phase in the Landau level plot. We had originally used the error analysis for this extrapolation as discussed in Schoenberg’s “Magnetic Oscillations in Metals”, but the referee’s suggestion is an excellent visual representation of the error so we have added it into Figure 3c. The error obtained from this analysis is similar, albeit slightly larger than what we previously reported. However, this does not take away from the main point which is that the Berry phase is definitively nonzero.

The authors claim that the SdH signal vanishes for films thicker than 2nm. Is this an abrupt effect? What happens for 2nm versus 2.4nm?

We have found that some of the effects that we report on change abruptly from 2.4 nm to 2 nm while others evolve gradually. For example, the amplitude of the low frequency SdH oscillations is strongly dependent on the thickness of the samples. The amplitude is largest in our 1.2 nm samples and decreases gradually until it is immeasurably small at 2.4 nm. Due to time constraints at the high magnetic field facility, we have not been able to check this thickness dependence of the high frequency oscillations. However, the anisotropic magnetoresistance and the shape of the magnetoresistance change abruptly from 2.4 nm to 2 nm. The AMR is almost immeasurable in 2.4 nm thick samples while it is very large in 2 nm and thinner samples. Additionally, the magnetoresistance shows an almost parabolic field dependence in 2.4 nm and thicker samples while samples that are at 2 nm and thinner exhibit a weak antilocalization correction to the magnetoconductivity as shown in the text.

Can the authors demonstrate that the SdH effect is 2D in nature? E.g. by tilting the field and observing a $1/\cos^2\theta$ dependence.

To probe the dimensional nature of the Shubnikov de Haas oscillations, we plan on including angular dependent magneto transport data at low and high fields in our follow-up study.

In line 160 I believe that it should be 1013.

We thank the reviewer for pointing this out and have fixed all the Hall effect carrier concentrations so we are confident they are now correct after the revision in the analysis described below.

Do I understand correctly that from the Hall and SdH analysis the authors obtain three bands: The first observed in low field SdH with 10^{11} , another one with 10^{12} both detected by the SdH as well and a third band that accommodates the majority of the carriers, which is not seen by SdH but dominates the Hall. How come the 10^{12} band is not observed in the Hall and the 10^{11} band is? The authors should clarify this point.

We apologize for the confusion in the Hall effect shown in the text. Unfortunately, there was a typo in our original analysis which used the wrong sample thickness when converting from the Hall voltage to the Hall resistivity. We have corrected this mistake, and the revised analysis shows that the low carrier band has a concentration of $1.9 \times 10^{12} \text{ cm}^{-2}$ which is consistent with the total carrier concentration observed in the oscillations. However, the Hall effect clearly shows another band which is not observed in the SdH oscillations and has been deduced from a two-band analysis of the non-linear Hall data to have a larger sheet carrier concentration of $1.9 \times 10^{14} \text{ cm}^{-2}$. This is probably because the mobility of this band is too low to observe oscillations even up to a 60 T field. We have added text to paragraph 1 on page 4 to clarify this.

The authors write “large positive in-plane AMR has been found in some [41, 42], but not all, samples.” Is it true that large and positive AMR is observed in polar oxide interfaces while it is absent in non-polar ones as suggested in ref. 42? This is in fact consistent with the current paper.

To our knowledge, there are no non-polar perovskite interfaces which show such a large AMR. This is indeed consistent with ref. 42 as the referee suggests. It is for this reason that we attribute the observed AMR to the Rashba effect. The polar interface should increase the strength of the Rashba coupling compared to a non-polar interface and therefore increase the size of the AMR. We have modified the text on paragraph 2 of page 5 to make this clearer.

Finally, how strong the obtained Rashba coupling comparing to the Ti atomic spin orbit coupling?

The atomic spin-orbit coupling constant for Ti^{3+} is about 88K (see for example Coey’s *Magnetism and Magnetic Materials* textbook) which is equivalent to 8meV. This is significantly larger than our Rashba spin orbit energy of about 1meV.

By addressing all the concerns of all four referees and making the appropriate major modifications to the manuscript, we believe that our manuscript is ready for consideration and publication in Nature Communications. Therefore, we sincerely request further consideration. We list in detail below the list of major modifications to the manuscript below.

Sincerely,

Yuri Suzuki

Major changes to the manuscript:

1. Modified paragraph 4 on page 1 to cite additional references, give a more in-depth background of transport measurements in LTO/STO heterostructures, and to clarify why we chose to study ultra-thin films. It now reads:

“LTO/STO heterostructures have previously been compared to the LAO/STO system, but there are a few differences which make these systems distinct from each other. In particular, when LTO is grown under large (-1.6%) compressive epitaxial strain on STO substrates, the films are metallic throughout the bulk of the film [15]. This metallicity is due to the fact that the epitaxial strain induces an electronic structure modification [20]. However there is also a contribution to conduction from electronic reconstruction at the LTO/STO interface which has a higher resistivity [16]. While the LAO/STO system has been much more widely studied, the magnetotransport of LTO/STO system has been well studied in films with thicknesses on the order of or greater than 10 unit cells. This includes electrical gating dependence of the electrical transport and analysis of the magnetoconductivity in the framework of weak antilocalization [21– 24]. However, in this thickness regime, our previous studies [15] have demonstrated that the electrical transport is dominated by the metallic conduction in the entire LTO film induced by the epitaxial strain. We have therefore chosen to study ultra-thin (3 unit cells) films of LTO grown on STO single crystal substrates where the metallic conduction is dominated by the interfacial contribution induced by the electronic reconstruction. These ultra-thin films exhibit properties drastically different from thicker films including the previously unobserved quantum oscillations and a nonlinear Hall effect under zero bias[21].”

2. Added paragraphs 1 through 4 on page 3 to include DFT calculations. These calculations are now used to justify our simple model in calculating the Rashba constant from the SdH oscillations. The paragraphs are the following:

“Complex oxide heterostructures often have complex band structures due to the contribution of multiple orbitals. So when analyzing SdH oscillations in these systems, it is necessary to take into account contribution from all of these orbitals. In our system, these orbitals include the $Ti\ d_{xy}$, d_{xz} , and d_{yz} orbitals.

There have been a number of theoretical modeling studies of the LTO/STO interface. Some of these studies have been performed in the context of superlattices where the interfaces cannot be considered to be isolated interfaces [2, 26] while others consider a single layer of LTO embedded in STO [27, 28]. However these studies do not necessarily shed light on the electronic structure for these ultra-thin LTO films which show quantum oscillations. Therefore we have theoretically modeled the electronic structure of 2 and 3 unit cells of LTO on STO using GGA+U calculations.

DFT calculations were used to calculate the effective masses of the xy , xz , and yz bands along the $\Gamma-X$ direction near the Γ point. The calculated masses are 0.70 m_e , 0.61 m_e , and 8.0 m_e respectively. If we naively attribute the higher frequency oscillations to the d_{xz} or d_{yz} bands because of their larger effective mass, then the effective mass would be the geometric mean of the mass along the $\Gamma - X$ and $\Gamma - Y$ directions (i.e. $m^ = \sqrt{m_x m_y}$), thus predicting an effective mass of 2.2 m_e . This value is considerably larger than the*

measured mass. So we cannot attribute either of the oscillations to d_{xz} or d_{yz} bands. Moreover, both of the measured masses are different from the calculated d_{xy} band. However, a Rashba effect would create an inner band with a smaller mass and an outer band with a larger mass than the unsplit band. So based on these effective masses, we propose that these quantum oscillations arise from a Rashba split d_{xy} band.

The DFT calculations also predicted the density of states near the Fermi level. Figure 4 shows the resulting density of states specifying the xz , yz , and xy character of the Ti 3d bands. These calculations clearly show that there is essentially no contribution from the xz and yz bands to the density of states near the Fermi level. Only parabolic xy bands that have nearly equal effective masses and are offset from each other in energy contribute to the conduction. Therefore, these calculations show that we can greatly simplify our analysis of the quantum oscillations by only considering parabolic d_{xy} bands split by a Rashba effect.”

3. Added figure 4 to show the band character of the density of states around the Fermi level.
4. Modified figure 3 to show the error analysis on the Berry phase as described by referee 4.
5. Modified paragraph 1 on page 4 to include more background on the nonlinear Hall effect in LTO/STO heterostructures, fixed the analysis of the concentrations, and described how the nonlinear Hall effect fits into our DFT calculations. The paragraph is now the following:

“A sheet carrier concentration of $(1.8 \pm 0.3)10^{11} \text{ cm}^{-2}$ was deduced from the low frequency oscillations and a sheet carrier concentration of $1.7 \times 10^{12} \text{ cm}^{-2}$ from the high frequency oscillations. Due to the non-linear field dependence of the Hall resistance at these low temperatures, as shown in Figure 1, a single carrier concentration value cannot be deduced from the Hall effect. This is in contrast to thicker films where the Hall effect is linear. The Hall data in the ultra-thin films can be fit to a two-carrier model using the zero field resistivity as an additional constraint [35]. Such a two-carrier model has previously been applied to electrically gated, thick (> 10 unit cells) LTO films on STO [21] and ungated δ -doped STO/LTO/STO superlattices [35]. In our case, we find a linear Hall effect in thick LTO films and a non-linear Hall effect for the thinnest of LTO films and only at low temperatures. Our analysis of ultra-thin LTO films at low temperatures gives estimates of two sheet carrier concentrations of $(1.9 \pm 0.2)10^{14} \text{ cm}^{-2}$ and $(2.0 \pm 0.2)10^{12} \text{ cm}^{-2}$ with mobilities of $46 \pm 1 \text{ cm}^2/\text{Vs}$ and $(4.0 \pm 0.1)10^3 \text{ cm}^2/\text{Vs}$ respectively. In the context of the DFT calculations, it appears that that these two carrier concentrations are associated with Ti d_{xy} bands from different layers in the heterostructure; one band likely comes from the Ti in the LTO film while the other comes from the Ti at the interface. In any case, the sum of the sheet concentrations deduced from the oscillations is equal to the low carrier concentration associated with high mobility carriers deduced from the Hall effect, thus indicating that the oscillations account for all of the high mobility carriers. The high concentration carriers deduced from the Hall effect appear to have too low of a mobility to observe any oscillations even up to 60 T.”

6. Modified paragraph 7 on page 4 to include the spin-orbit scattering time. The added sentence is the following:
“A value of 3×10^{-3} ps was obtained for the spin orbit scattering time which is significantly smaller than the typical spin orbit scattering times of 1 ps obtained for LAO/STO [5].”

7. Added a paragraph in the “Methods” section for the DFT calculations which reads:
“Density functional theory (DFT) calculations were performed using the full-potential linearized augmented plane wave (FP-LAPW) code WIEN2k [66]. The exchange-correlation potential was calculated with the generalized gradient approximation (GGA) [67] and the fully localized and rotationally invariant “+U” correction [68,69]. The U_{eff} (=U-J) parameter has been set to 5 eV and 7 eV, respectively for the 3d electrons of Ti atoms, and the 4f electrons of La atoms. A c (2x2) lateral unit cell was used containing two non-equivalent Ti sites per atomic layer was used in order to take into account of possible structural distortions, with a lateral lattice parameter set to the GGA equilibrium lattice constant of STO (3.92 Å, slightly larger than the experimental value 3.905 Å). The atomic positions were fully relaxed. In order to avoid the emergence of a spurious electric field due to the periodic boundary conditions, we have chosen a LTO(3MLs)/STO(1.5MLs)/LTO(3MLs)(001) symmetric slab with LTO layers on both sides of the STO substrate and a vacuum region between the slab and its periodic images of 20 Å.”

8. Added R. Arras and R. Pentachora as authors for their contributions of the DFT calculations.

Reviewers' comments:

Reviewer #1 (Remarks to the Author):

Also after the rebuttal letter, I hold that the physical interpretation of the quantum oscillations is almost certainly incorrect. The quantum oscillations experiments indicate two different Fermi surfaces, and the authors interpret these two Fermi surfaces as having one xy orbital which is split by the Rashba effect into two. This interpretation necessitates an extraordinarily large Rashba splitting, the largest one ever reported in oxides.

The much more likely interpretation is that the two Fermi surfaces originate from two Fermi surfaces that have an other origin. Possible candidates are, as I wrote in my first report, (i) Fermi surfaces originating from different LTO and STO layers at the interface (ii) different orbitals xy vs. xz/yz , and (iii) several copies of the xy orbital because of the quantum confinement in three layers.

All of (i)-(iii) can lead to largely different Fermi areas including very small ones, and can explain the two SdH oscillations without the need of an extraordinarily large Rashba splitting.

To rebut point (ii) the authors have carried out GGA+U calculations and report a single xy orbital and no xz/yz orbitals at the interface. However, if U is large enough (and the authors use a pretty large one) GGA+U will always result in such a large orbital polarization that only a single orbital remains at the Fermi surface for every material (with occupations $n \leq 1$ electrons per site). DFT and DFT+DMFT calculations which are not biased in this respect do show both xy and xz orbitals at the Fermi surface, see PRB 87, 241101(R). Also experimentally xz/yz orbitals are actually found in ARPES experiments for various LTO/STO heterostructures, see PRL 111, 126401 (2013).

As for (iii), the authors say that "our quantum oscillations are vastly different from each other and therefore we cannot be attributed to two different xy bands." This is not correct since the xy bands are shifted so that with the same mass the Fermi surface area can be largely different.

The authors do not reply to point (i). Again the xy orbital in the LTO and STO interface layers can have largely different areas. Actually on p. 34 the authors use this scenario of different xy bands to interpret their Hall resistance in a two carrier model. This can simply give rise to two Fermi surface areas and hence frequencies, without the need to rely on an extraordinarily large Rashba splitting. This interpretation of the Hall resistance is an

internal contradiction within the paper: (i) the GGA+U does not show different xy bands at different layers of the interface and (ii) the different layers should also give different oscillation frequencies.

In conclusion, there are more natural ways to explain very different Fermi surface areas and hence oscillation frequencies, without the no need of an extraordinarily large Rashba splitting. ARPES experiments [PRL 111, 126401 (2013)], theory [PRB 87, 241101(R)] and the authors own interpretation of the Hall resistance all point in this direction. Extraordinary claims require at least some reasonable evidence which the authors do not provide. Hence the authors interpretation is almost certainly incorrect.

Reviewer #2 (Remarks to the Author):

Significant efforts have been made by the authors to address the referees' comments, as evident from inclusion of the results of electronic structure calculations, and several new references relevant to electronic structure and electron transport in LaTiO₃/SrTiO₃ epitaxial structures. On the basis of these modifications and clarifications, I consider this manuscript suitable for publication in the Nature Communications as a regular article.

In passing, however, it is worthwhile to add that if epitaxial strain induced metallicity as predicted in Phys. Rev. B 77, 115350 (Ref. # 20 of this manuscript) is indeed significant, it should be most prominent in the thinnest films as the strain would relax on increasing the film thickness through creation of defects. Such strain-free films should be in the Mott insulating state unless there is excess oxygen to make them conducting. Although it may be a daunting task to separate the contributions of epitaxial strain and electronic reconstruction at the interface to metallicity of a few unit cells thick films, it surely appears a relevant problem brought forward by this manuscript, which in my opinion deserves publication in its present revised form.

Reviewer #3 (Remarks to the Author):

The revised version of the manuscript "Quantum oscillations resulting from a giant Rashba spin splitting in LaTiO₃/SrTiO₃ heterostructures", by M. J. Veit, et al., has been improved to some extent. The absence of superconductivity in the given sample may well be consistent with the fact that xz and yz bands should be filled for superconductivity to appear. There are two aspects that puzzle me, though. First of all, the xy, xz and yz orbitals should be degenerate in the bulk. As far as I can understand, the main effect lifting this degeneracy should be quantum

confinement near the interface (in a roughly triangular potential well, whose slope gives the electric field at the interface). Now, xz and yz bands have a lighter mass in the direction perpendicular to the interface, as compared to xy bands, so my expectation was that these bands should be found HIGHER than the xy bands. Can the author explain the physical mechanism pushing them DOWN? The second puzzling aspect is somewhat related. The authors claim that the system exhibits a GIANT Rashba spin splitting. This should be associated with a large electric field at the interface (and I am perfectly fine with this). The questions are: what is the effect of such a large electric field on the DFT calculations? How is it taken into account? Is it evident that the outcome of DFT calculations can be equipped (by hand) with a large Rashba coupling, without contradictions?

I suggest that the authors should improve the discussion on the above aspects, before their piece of work is reconsidered for publication in Nature Communications.

Reviewer #4 (Remarks to the Author):

I read the revised manuscript and the detailed response letter. I have received only an edited version of the other Referees' comments. It was therefore somewhat difficult to understand what were their exact remarks.

Nonetheless, the paper has improved significantly and the DFT simulation certainly added to the understanding of this interface. I still think that the paper could have been more complete had the gate and angular dependence been included. Gate dependence is important to demonstrate that the spin-orbit effect is indeed a Rashba-type and that it is tunable, a property that may be important for future spintronics applications. A frequency that has a $1/\cos(\theta)$ dependence, with θ being the angle between the perpendicular to the surface and the magnetic field would prove a 2D behavior, ruling out bulk effects from a possibly oxygen deficient STO crystal. Since the authors do not have such data at hand I do not suggest holding the communication of the data and recommend publication at the present form.

Point-by-point response to reviewer comments

REVIEWER #1

[Here we address the points made by reviewer #1 in the order of the possible scenarios that he/she has listed. All of the reviewer #1 comments from the second round of reviews are italicized and in blue for clarity.]

Also after the rebuttal letter, I hold that the physical interpretation of the quantum oscillations is almost certainly incorrect. The quantum oscillations experiments indicate two different Fermi surfaces, and the authors interpret these two Fermi surfaces as having one xy orbital which is split by the Rashba effect into two. This interpretation necessitates an extraordinarily large Rashba splitting, the largest one ever reported in oxides.

We point out here that the reviewer only focuses on the observation of the two sets of quantum oscillations throughout his first and second round of reviews and ignores the most important observation of a Berry's phase of π that has never been observed in a complex oxide interface system. In addition to the Berry's phase of π , we also observe a large anisotropic magnetoresistance (AMR) (larger than any observed in other complex oxide interface systems) and a weak anti-localization correction to the magnetoconductivity that are consistent with a giant Rashba splitting. The Rashba constant extracted from the weak anti-localization fit to the magnetoresistance even matches what was extracted from our model of the quantum oscillations. If we only had observed two sets of quantum oscillations, the large AMR and a weak anti-localization correction to the magnetoconductivity, the case for Rashba splitting may not be as definitive. However, the Berry's phase of π cannot be explained by the reviewer's proposed alternatives, and a Rashba splitting is the simplest and most natural explanation for all of the data.

We have done a more thorough analysis of the character of all the bands in the band structure derived from DFT calculations and have determined that the bands closest to the Fermi energy are an xy band and a hybridized $xz+yz$ band. We had originally interpreted this hybridized band to be an xy band based on the band's isotropy and looking at the total density of states. Contrary to our original interpretation, the isotropic nature of the band actually comes from the hybridization of the xz and yz orbitals. While this hybridized band is almost completely isotropic near the Γ point, the shape of the Fermi pocket is not exactly circular. This makes our previous model of a parabolic band more inaccurate. So, we now model this band with a tight binding model that includes a linear Rashba effect as described on page 4 paragraph 1. We chose to model this band with a tight binding model as opposed using the DFT calculations directly so that we can get high resolution near the Γ point where the Fermi pockets from the oscillations are. This change has very little effect on the interpretation of a giant Rashba splitting though and there is only a minor change in the Rashba constant fit from this model.

The much more likely interpretation is that the two Fermi surfaces originate from two Fermi surfaces that have another origin. Possible candidates are, as I wrote in my first report, (i) Fermi surfaces originating from different LTO and STO layers at the interface (ii) different orbitals xy vs. xz/yz , and (iii) several copies of the xy orbital because of the quantum

confinement in three layers. All of (i)-(iii) can lead to largely different Fermi areas including very small ones, and can explain the two SdH oscillations without the need of an extraordinarily large Rashba splitting.

We now address below point by point the detailed comments that reviewer #1 has made associated with the above three scenarios that do not take the Berry's phase into account.

The authors do not reply to point (i). Again the xy orbital in the LTO and STO interface layers can have largely different areas. Actually on p. 34 the authors use this scenario of different xy bands to interpret their Hall resistance in a two carrier model. This can simply give rise to two Fermi surface areas and hence frequencies, without the need to rely on an extraordinarily large Rashba splitting. This interpretation of the Hall resistance is an internal contradiction within the paper: (i) the GGA+U does not show different xy bands at different layers of the interface and (ii) the different layers should also give different oscillation frequencies.

We disagree with the reviewer that we did not reply to point (i) which states that the quantum oscillations are possibly due to the “Fermi surfaces originating from different LTO and STO layers near the interface.” In fact, the new DFT calculations that we had included in the revised manuscript from April 2017 describe the contributions of each of the LaTiO₃ and SrTiO₃ layers near the interface. We previously plotted the total density of states in Figure 4 of the main text (note that Figure 4 has now been changed, but the information in the original figure can be found in supplementary Figures S6 and S7) but had clearly included the contribution from xy orbitals from each of the layers in Figure S6 of the supplemental section and from the xz and yz orbitals from each of the layers in Figure S7 of the supplemental section.

In order to make it clearer exactly which bands are contributing to conduction, we now include the entire band structure from the DFT calculations. We had originally thought that just showing the density of states would be the best way of showing the results, but there was clearly some confusion so this was not the case. The new figure shows that there are 2 bands that cross the Fermi energy as the reviewer suggests. One band indeed crosses the Fermi energy close enough to the Γ point to have a small Fermi surface area like we measure. However, the other band has a Fermi surface area that is far too large to explain even our high frequency oscillations.

The contributions from the interface and the SrTiO₃ layer are the two sources of carriers that contribute to the Hall resistance measurements: lower concentration/ higher mobility and higher concentration/ lower mobility carriers. The lower concentration high mobility carriers ($2.0 \times 10^{12} \text{ cm}^{-2}$ and $4400 \text{ cm}^2/\text{Vs}$ from the Hall resistance) are comprised of the Rashba split xz+yz hybridized band in the SrTiO₃ layer (band 1 in the new figure 4) with two Fermi pockets of $1.8 \times 10^{11} \text{ cm}^{-2}$ and $1.7 \times 10^{12} \text{ cm}^{-2}$ (as measured from the quantum oscillations) adding up to the total sheet carrier concentration obtained from the Hall data. The higher concentration/low mobility carriers correspond to carriers from the interface layer xy band according to DFT calculations (band 2 in the new figure 4) and it is likely that these low mobility carriers do not exhibit oscillations due to their shorter mean free path.

The reviewer's statement that our interpretation of the "Hall resistance in a two carrier model" is "an internal contradiction" is therefore not correct. Firstly, the reviewer states that "GGA+U does not show different xy bands at different layers of the interface." This is not what the DFT calculations show in Figure S6, and the new figure 4 shows clearly the contributions from different layers. Secondly the reviewer asserts that the different layers should give different oscillation frequencies. While this is indeed true, in our case, the low mobility carriers from the xy orbitals at the interface should have a frequency that is orders of magnitude higher than what we observe. Based on the Hall effect fit, the frequency from this band should be on the order of 3000 T. Moreover, the fact that these lower mobility carriers do not exhibit oscillations is not surprising because high mobilities are a necessary requirement to observe quantum oscillations at such low fields.

Based on the reviewer's comments, it is clear that we did not properly explain the two-carrier fit in terms of our model. Indeed, there should actually be three different carrier types in the Hall effect measurements. One comes from the xy band at the interface and two come from the Rashba-split $xz+yz$ hybridized band in the SrTiO_3 layer. However, the mobilities of the carriers from the SrTiO_3 layer should be similar which makes it impossible to accurately determine all of the carrier concentrations from a three-band model of the Hall effect. We therefore fit the Hall effect to a two-carrier model to obtain the carrier concentration from the interface xy band and the sum of the carriers from the Rashba split band in the SrTiO_3 . Therefore, what the reviewer claims to be an internal contradiction is not in fact a contradiction but can be easily explained.

We have now rewritten the discussion of the low and high concentration carriers from the Hall effect measurements on page 4 to clarify that the high mobility low concentration carriers that give rise to the two sets of oscillations are from the $xz+yz$ orbitals of the SrTiO_3 and that the low mobility high carrier concentration carriers are from the xy orbitals of the interface layer according to DFT calculations. We have also updated figure 4 to show the entire band structure instead of the density of states to make this explanation even more clear.

To rebut point (ii) the authors have carried out GGA+U calculations and report a single xy orbital and no xz/yz orbitals at the interface. However, if U is large enough (and the authors use a pretty large one) GGA+U will _always_ result in such a large orbital polarization that only a single orbital remains at the Fermi surface for _every_ material (with occupations $n \leq 1$ electrons per site). DFT and DFT+DMFT calculations which are not biased in this respect do show both xy and xz orbitals at the Fermi surface, see PRB 87, 241101(R). Also experimentally xz/yz orbitals are actually found in ARPES experiments for various LTO/STO heterostructures, see PRL 111, 126401 (2013).

As discussed above, the DFT calculations do find both dx_y and $xz+yz$ orbitals originating from the interface or close to it, thus our findings are fully consistent with the expectation of the referee. First of all, we would like to point out that the DFT/DMFT calculations in PRB 87 241141 and the DFT calculations that the reviewer #1 cited in the first round of reviews (J. Phys. Cond. Matt. 22 043001 (2010)) are theoretical studies focused on short period

superlattices of LaTiO₃ and SrTiO₃. While these studies may be compared with one another and with experimental studies on these kinds of superlattices, our bilayer system cannot be compared with either of these theoretical studies. Moreover, most of the DMFT studies so far have considered a single Ti site per layer which does not allow for more complex electronic reconstructions and, in this sense, represents a significant constraint. The large system size of this configuration that we model preclude the use of methods such as DFT+DMFT. Some of us have in the past performed DFT calculations on LaTiO₃/SrTiO₃ superlattices (PRL 07, JPCM 2010) but this model is not applicable to describe a thin LTO film on STO (001). The U value of 5eV used in the cited PRB reference above is similar to our U values of 5-7 eV. Therefore, differences in U values should not be the source of discrepancy between our DFT calculations and those of the cited PRB reference.

The reviewer also cites that “experimentally xz/yz orbitals are actually found in ARPES experiments for various LTO/STO heterostructures.” A closer look at the cited ARPES data shows that the anisotropic xz/yz bands are visible only when there is at least one layer of SrTiO₃ on top of the unit cell of LaTiO₃. The xz/yz bands then become more pronounced as more SrTiO₃ layers are added. This observation suggests that the observed xz/yz contribution is from the top SrTiO₃ layer and not the interface. In fact, in the sample with the thickest SrTiO₃ cap of 3 unit cells, ARPES likely only penetrates the SrTiO₃ layer as it is a surface sensitive technique. Therefore, these ARPES measurements only conclusively indicate that pure xz/yz orbitals should exist in the SrTiO₃ capping layer. Our DFT calculations would suggest that hybridization of the xz and yz orbitals would give rise to isotropic bands but these predictions need to be confirmed in uncapped LaTiO₃ films.

As for (iii), the authors say that "our quantum oscillations are vastly different from each other and therefore we cannot be attributed to two different xy bands." This is not correct since the xy bands are shifted so that with the same mass the Fermi surface area can be largely different.

In reviewer #1's statement above, he/she implies that the two sets of oscillations can be explained by shifted xy bands with the same mass. This is simply not true and indicates that reviewer #1 has not read through paragraph 4 page 2 of our manuscript. Indeed, one of the major results of our paper is that the two sets of oscillations exhibit effective masses which differ by an order of magnitude: 1.2 m_e and 0.12 m_e. Such different effective masses cannot be derived from shifted xy bands.

REVIEWER #2

We thank reviewer #2 for his/her positive recommendation for publication in its present form. We address below the one comment that he/she raised.

In passing, however, it is worthwhile to add that if epitaxial strain induced metallicity as predicted in Phys. Rev. B 77, 115350 (Ref. # 20 of this manuscript) is indeed significant, it should be most prominent in the thinnest films as the strain would relax on increasing the film thickness through creation of defects. Such strain-free films should be in the Mott insulating state unless there is excess oxygen to make them conducting. Although it may be a daunting task to separate the contributions of epitaxial strain and electronic reconstruction at the interface to

metallicity of a few unit cells thick films, it surely appears a relevant problem brought forward by this manuscript, which in my opinion deserves publication in its present revised form.

We appreciate the reviewer's comment that indeed epitaxial strain induced by metallicity is an important ingredient in three unit cells of LaTiO_3 on SrTiO_3 . In our previous study that the reviewer cites and more recent films that we have synthesized in our lab, we do see a scaling of the resistance with thickness from 2nm up to 60nm, thus indicating that strain relaxation effects are not significant up to this thickness value. This is consistent with reciprocal space maps which show very little relaxation in films up to 60 nm thick. In any case, when strain relaxation begins to occur, the strain relaxed Mott insulating portions of the sample would likely be shorted out by more metallic parts of the sample.

In our 3-unit cell (1.2nm) thick films, we find that the sample does not scale with the resistance of thicker films, thus suggesting that the resistance associated with the $\text{LaTiO}_3/\text{SrTiO}_3$ interface dominates conduction. Separating out the contributions of the epitaxial strain from the electronic reconstruction would be "a daunting task" as the reviewer states. A more detailed study of 1 (0.4nm) to 5 (2nm) unit cell thick LaTiO_3 films on SrTiO_3 would be required. The accuracy with which we can grow a contiguous unit cell thick LaTiO_3 film on SrTiO_3 may pose a significant challenge but may be overcome with careful synthesis.

REVIEWER #3

We thank reviewer #3 for his/her careful reading of the manuscript, especially the newly added portion regarding the DFT calculations. We appreciate his/her assessment that the manuscript has been "improved to some extent." We address below the two points that the reviewer states are puzzling to him/her.

First of all, the xy, xz and yz orbitals should be degenerate in the bulk. As far as I can understand, the main effect lifting this degeneracy should be quantum confinement near the interface (in a roughly triangular potential well, whose slope gives the electric field at the interface). Now, xz and yz bands have a lighter mass in the direction perpendicular to the interface, as compared to xy bands, so my expectation was that these bands should be found HIGHER than the xy bands. Can the author explain the physical mechanism pushing them DOWN?

We agree with the reviewer that quantum confinement near the interface lifts the degeneracy of the xy, xz and yz orbitals. What appears to be the xz and yz bands shifting down as the reviewer suggests is actually a gap opening because of the on-site Coulomb repulsion term. This is clear when looking at energies above the Fermi level because there is a restoration of the density of states for the xz and yz bands. The reason why the xz and yz bands show a gap while the xy bands do not is because the quantum confinement greatly reduces the hopping integral for the xz and yz orbitals while leaving the xy orbital hopping integral relatively unchanged. So, the Coulomb repulsion will shift the occupied xz and yz bands lower in energy due to a larger Mott gap while leaving the xy band unchanged. So, we agree that, for a given layer, quantum confinement will cause the xz and yz bands to be higher in energy than the xy band, but the

introduction of Coulomb repulsion and multiple layers causes some xz and yz bands to be lower than some xy bands as is indicated in the band structure in the updated Figure 4.

The second puzzling aspect is somewhat related. The authors claim that the system exhibits a GIANT Rashba spin splitting. This should be associated with a large electric field at the interface (and I am perfectly fine with this). The questions are: what is the effect of such a large electric field on the DFT calculations? How is it taken into account? Is it evident that the outcome of DFT calculations can be equipped (by hand) with a large Rashba coupling, without contradictions?

Due to the polar discontinuity both at the LTO surface and at the LaTiO₃/SrTiO₃ interface, the internal electric field is intrinsic to the system, and is naturally present in the calculation (we note that in order to avoid artifacts we model two surfaces but within each half of the simulation cell there is an electric dipole). Similar observations of an inherent field are made in the LaAlO₃/SrTiO₃ system in references such as reference 29. Therefore, the DFT calculations were not equipped (“by hand” as the reviewer suggests) with a large electric field and there is no inherent contradiction in the results. When we fit the oscillations to a Rashba split parabolic band, what is added is actually spin-orbit coupling rather than an internal electric field. Spin-orbit coupling only has a very small effect on the overall band structure, and it does not take away from the main point of the calculations which is determining the bands that cross the Fermi level.

We have changed our model for fitting the quantum oscillations from a simple parabolic model to a tight binding model to more accurately capture the features of the DFT calculations. The reason that we used a tight binding model with a Rashba constant to include the spin-orbit coupling is that it provides us better resolution near the Γ point. Because the Fermi surface areas from the quantum oscillations are so small, we require a very fine resolution near the Γ point, but the step size in k-space of the DFT calculations is too large to accurately capture these effects when spin-orbit coupling is added. Therefore, there are again no contradictions between the DFT calculations and the model for fitting the oscillations.

REVIEWER #4

We thank the referee for his/her recommendation for publication in its present form. We do appreciate his/her comments about measuring the gate dependence and the angular dependence of the magnetotransport measurements. Fabrication is currently underway to both back gate and top gate these samples to modulate the carrier concentration. We will also apply for high magnetic field time at the pulsed field facility in Los Alamos during the next proposal cycle in order to perform both gating and angular dependent magnetotransport measurements.

Below we list the major changes made to the manuscript in response to the second round of reviews that are addressed above.

Major Changes to the Manuscript

1. Updated the title to be “*Nontrivial Berry Phase in Quantum Oscillations Resulting from a Giant Rashba Spin Splitting in LaTiO₃/SrTiO₃ Heterostructures*” to emphasize the Berry phase result.
2. The parabolic model for fitting the Rashba constant was replaced with a tight binding model, and the new Rashba constant has been updated to 1.8×10^{-11} eVm.
3. Changed Figure 4 from the density of states to the DFT band structure.
4. Changed the Figure 4 text so that it corresponds to the band structure to now read “*DFT+U results for the projected band structure for 3 monolayers of LTO on STO (001). The blue and orange bands (labeled 1 and 2 respectively) are close enough to the Fermi level to contribute to the electrical transport. The black bands are too far away to contribute any carriers.*”
5. Modified page 3 paragraph 2 to “*DFT calculations shed insight into the bands that are relevant to conduction. Figure 4 shows the calculated band structure. There are clearly only two bands crossing the Fermi level that contribute to the electrical transport (bands 1 and 2 shown in blue and orange respectively in Figure 4). Because the frequencies of the observed oscillations are very small, they cannot be attributed to the Fermi pocket created by band 2. DFT calculations indicate that band 2 comes from the interfacial d_{xy} orbitals while band 1 comes from the d_{xz} and d_{yz} orbitals in the STO. Interestingly, band 1 is actually a hybridization of d_{xz} and d_{yz} orbitals rather than purely one or the other as is usually expected in STO. While d_{xz} and d_{yz} bands are typically highly anisotropic, this hybridization causes the band to be essentially isotropic near the Γ point. Hybridization of the bands may be caused by distortions of the nearly cubic STO lattice in the presence of electric field from the polar discontinuity [29]. In any case, both bands crossing the Fermi level appear to be isotropic.*”
6. Modified page 3 paragraph 3 to include the sentence “*The effective mass of the hybridized band is nearly the same as the d_{xy} bands near the Γ point, but it increases at higher wave vectors due to an anisotropy which is not apparent at low wave vectors.*”
7. Modified page 4 paragraph 3 to include “*There are three carrier types from two different bands in total which contribute to conduction. One band from the d_{xy} orbitals on the interfacial Ti contributes the low mobility/high concentration carriers (band 2 in Figure 4). The other two carriers come from the d_{xz+yz} band in the STO (band 1 in Figure 4), and we attribute the appearance of two Fermi pockets to a Rashba splitting from a single band in the STO. This band contributes the high mobility/low concentration carriers observed in the Hall effect and both of the Fermi pockets observed in the quantum oscillations.*”
8. Added the following paragraph as paragraph 4 on page 3: “*The band structure in Figure 4 shows that the d_{xy} band (band 2) crosses the Fermi energy at approximately $X/2$ which is much larger than what is measured based on the frequencies of either of the oscillations. Therefore, the d_{xy} band cannot give rise to the Fermi pockets measured by the quantum oscillations. On the other hand, the hybridized d_{xz+yz} band (band 1) is near the Fermi energy near the Γ point so it can give rise to such a small Fermi pocket. Because two sets of oscillations are observed with only one band in the band structure that can produce a Fermi pocket small enough to account for the frequencies, the simplest and most likely explanation is a spin-splitting of this hybridized d_{xz+yz} band.*”

9. Modified page 4 paragraph 1 to now include the new tight binding model. It now includes the sentences *“In order to properly model the properties of the hybridized d_{xz+yz} band, we extended the tight binding model described by Rodel et al [32] to include a linear Rashba splitting. Using this model, the energies of the resulting bands are given by $E(k)=\varepsilon_{xz+yz}(k)\pm\alpha|k|$, where α is the Rashba coupling constant and $\varepsilon_{xz+yz}(k)$ is the wave vector dependence of the energy without spin-orbit coupling. The hopping integrals of the tight binding model were fitted to the DFT bands to determine $\varepsilon_{xz+yz}(k)$. Then, the Rashba coupling constant was adjusted to fit the effective masses and Fermi surface areas measured by the quantum oscillations.”*
10. Added the following sentence to page 4 paragraph 2: *“It has been suggested that high mobility carriers are associated with the filling of d_{xz} and d_{yz} bands [38] and our combined experimental and DFT results are consistent with these previous findings.”*
11. Added the following paragraph on page 4 as paragraph 3: *“Together our experimental and theoretical results suggest that the different carriers come from d_{xy} and d_{xz+yz} hybridized bands associated with different parts of the heterostructure. There are three carrier types in total from two different bands which contribute to conduction. One band from the d_{xy} orbitals on the interfacial Ti contributes the low mobility/high concentration carriers observed in the Hall effect (band 2 in Figure 4). The other two types of carriers come from the hybridized d_{xz+yz} band in the STO (band 1 in Figure 4) that is Rashba split into an inner and outer Fermi surface. This band contributes the high mobility/low concentration carriers observed in the Hall effect and both of the Fermi pockets observed in the quantum oscillations.”*
12. Modified page 6 paragraph 1 to include the sentences *“There are several possible scenarios which can explain multiple quantum oscillations in these complex oxide heterostructures. First the oscillation could come from an d_{xy} band while the other comes from a pure d_{xz} or d_{yz} band. However, as explained previously, the effective mass of the low frequency oscillations is much smaller than what we would expect for a pure d_{xz} or d_{yz} and this situation does not explain the nontrivial Berry phase. Second, the oscillations could come from d_{xy} bands from different LTO and STO layers. However, in this scenario the effective masses of the oscillations should be the same which is not what we observe, and again this does not explain the Berry phase. A third scenario to explain this data would be topological edge states. This would indeed explain the Berry phase, the AMR, and the weak anti-localization. However, we have found no theoretical justification to make such a claim. The fourth possible explanation is a giant Rashba effect which is the simplest explanation consistent with all the data presented here.”*

Reviewers' comments:

Reviewer #3 (Remarks to the Author):

The revised version of the manuscript “Nontrivial Berry phase in quantum oscillations resulting from a giant Rashba spin splitting in LaTiO₃/SrTiO₃ heterostructures”, by M. J. Voit, et al., appears to have been improved with the introduction of significant changes. I agree with the present interpretation of the DFT results, and with the identification of the minority high mobility carriers with electrons filling dxz/dyz (hybridised) sub-bands, and of the majority low mobility carriers with electrons filling dxy sub-bands. By the way, this identification seems to have already been explicitly put forward in Ref. 38, if not even before. I am still dissatisfied with the discussion about the relation between the giant Rashba spin splitting and DFT calculations, but this comes rather to my fault, as I was not able to make myself understood in my previous report. I shall try to make my point clearer: given that the large electric field arising at the interface is captured by their DFT calculations, do the authors expect that any reasonable treatment of spin-orbit coupling within DFT may result in a giant Rashba-like spin splitting comparable to the one they deduce from the measurements? My wording referring to a field that must be “introduced by hand” was not meant to refer to the lack of polar discontinuity in the DFT results (I am sure that this effect is captured), rather to the possibility to quantitatively recover the giant Rashba-like spin splitting WITHIN the same DFT scheme, once (atomic) spin-orbit coupling is taken into account. My attitude is rather simple-minded here, but I am pretty sure the reader may have the same curiosity: if the authors had been able to obtain a reasonable estimate of the Rashba-like spin splitting within DFT, they would present their results in support of their data. The fact they did not, seems to indicate that this was not possible, and the reader may well wonder why. I ask the authors to discuss this aspect, which may be of interest to an even broader readership.

Besides this clarification, I think that, although not fully conclusive as far as the identification of the relevant electron states is concerned, the present work is rather interesting and sound, and will certainly trigger new experimental and theoretical investigation on oxide heterostructures.

Once the authors have considered my suggestions, I think that their piece of work may be accepted for publication in Nature Communications.

Reviewer #4 (Remarks to the Author):

I agree with Reviewer #1 that there is some inconsistency between the Hall data and the Shubnikov-de Haas measurements as I mentioned in my first review. The π Berry phase is

indeed interesting assuming that the error on this phase is estimated correctly. I would leave these questions to the readers of the paper. I recommend publication.

Reviewer #5 (Remarks to the Author):

The manuscript by M.J. Veit et al. deals with the observation of a large Rashba-type spin splitting at the interface of thin LaTiO₃ films on SrTiO₃ by magnetotransport experiments. I have read the full manuscript file and all previous referee comments and the response of the authors to those comments.

The authors have performed a systematic study and give enough background information to check the validity of their claims. I also appreciate that they tone down their claims and use language along the lines that the results are “consistent with a large Rashba splitting”. Indeed all the observations are consistent with such an interpretation, but don’t provide a definitive proof. As a result, reviewer #1 draws into question the physical interpretation of the results and suggests that there might be another explanation. The main reason for doubting the interpretation is that the found Rashba-type spin splitting should be “extraordinary large” and the “largest one ever reported in oxides.” Also reviewer #3 is surprised by the magnitude of the found Rashba splitting, and the authors themselves also state that the splitting is “surprising”. It appears to me that reviewer #1 would be willing to accept the results if the found splitting appears to be reasonable.

In my opinion the reason for this surprise about the magnitude of the Rashba-type spin splitting is due to the rather simplistic model used by both authors and reviewers with regard to the origin of the Rashba-type spin splitting. In this model a electric field is responsible for the size of the spin splitting, which is of course the original interpretation for semiconductor heterostructures. However, the study of Rashba-type effects at surfaces and also bulk systems has lead the field to the understanding that such fields play only a very minor role and that the main contributions are of a structural nature. A buckled atomic structure will result in a extreme enhancement of the spin splitting. In J. Phys.: Condens. Matter 21 (2009) 403001 a review of this argumentation can be found.

The polar nature of STO and LTO in combination with relaxation at the interface can result in a large atomic corrugation and thus Rashba effect. For the STO to vacuum interface this results in a Rashba constant of 5×10^{-11} eVnm (Nature Materials, 13, 1085 (2014)), more than 2 times as large as found here. These results are still being debated and the spin splitting found in the present manuscript can help shed a light on this. In a general sense one can thus state that the found Rashba-type spin splitting is not surprisingly large, but along the lines of what can be expected for a strongly localized 2D state at a buckled interface. Inclusion of a discussion along these lines will enhance the quality of the manuscript even further and should also convince reviewer #1 that the Rashba splitting is indeed the simplest explanation, also given the Berry

phase.

In my opinion the manuscript is suitable for Nature Communications and can be accepted for publication if the following changes are made:

- The authors should remove the claim of the largest Rashba-type spin splitting in oxides and include a reference and discussion of the Nature Materials paper mentioned above.
- The authors should revise their discussion of the Rashba effect to include structural effects. Furthermore, under these considerations it is technically no longer correct to refer to it as Rashba splitting, but typically one uses Rashba-type spin splitting in literature.
- The discussion and use of terminology of the effective mass is rather confusing. In a first approximation, the Rashba effect should not influence the effective mass when referring to the curvature. In line 244 the authors make the link to the effective mass as determined from their experiments and I agree with this statement, but the statement in line 205 can be misinterpreted and should be rephrased for clarity.
- It would be helpful if the authors show a bandstructure around Gamma as obtained from their tight-binding analysis including the correct energy and momentum scale. This will make it easier to compare the results to other studies.
- (minor) The states in the seminal ref. 33 actually are the surface states of BiTeI, as was found out in later studies (PRL 109, 116403 (2012) and PRL 109, 096803 (2012)). The bulk Rashba constant was determined in PRL 109, 116403 (2012).

Reviewer #6 (Remarks to the Author):

After revising the manuscript and the correspondence with Referees, I'm commenting in the following the main aspects that, in my opinion, remain disputed.

1) Referee #1 argued that bands arising from LTO and STO rather than Rashba-induced band splitting could be relevant for the observed two different oscillations. In their answer, the authors disagree with the referee opinion and included a new plot (Fig 4) intended to convince referee (and potential readers) that the xy band (labelled 2) and zz+yz bands (labelled 1) crossing the Fermi level, are of STO parentage. Fig. 4 only shows that a band (labelled 2) crosses the Fermi level, and a band (labelled 1) slightly touches the Fermi level. Unfortunately, this plot does not allow visualizing easily the parentage of the corresponding orbitals. This is better seen in Figs. S6 and S7 in Supplementary information. Whereas Fig S6 shows that interfacial xy orbital dominate the Fermi level, a smaller contribution of xz+yz of neighboring STO and LNO layers also exist (Fig S7). Although the STO-derived density of states is somewhat higher than that of LTO the difference seems to be rather marginal. If so, STO and LTO carriers will participate to conduction, probably with different effective masses and mobilities. It seems that authors do not appear to consider the contribution of LTO subbands. This is somewhat surprising as authors are aware (Ref 16) that ultrathin (strained) LTO films on STO are metallic

2) Carrier concentration and mobility are extracted from Hall data and used to support the claim that there are two sources of carriers. This method is much used but results are not always reliable. Basically, the problem arises because the Hall voltage is a very smooth function and the extracted parameters are much correlated. In fact, inspecting the extracted values for each transport channel, it turns out the corresponding carrier density and mobility values are different but, suspiciously, their product (carrier density x mobility) ($\approx 8-9 \cdot 10^{15}$) is almost identical for both channels. In this context, it is interesting to note that in Supplementary Infor (III) where the fitting of Hall data is described, the values of the carrier density and mobility for each band are different than those in the body of the manuscript. This may illustrate the fact that the fitting is not unique and thus the extracted values and their claimed agreement with data obtained from ShH oscillations may not be fully significant.

3) A value of the Berry phase is deduced from the observed oscillations of $R(H)$. The relevance of the extracted value, assuming it is meaningful, is not much discussed. However, it is emphasized in a rather mysterious way (“nontrivial Berry phase ...”) in the title. I’m not sure that the title is indicative of the strength of the manuscript. In fact, it seems that the authors only use the extracted value (π) as a potential support to the presence of Rashba splitting.

4) Referee #2 raises a crucial question regarding material’s issue. The point is that LTO thin films (< 10 uc) are metallic if grown on STO (ref 15 and 16, by authors). In reference 15 it was first argued that metallicity comes from strain effects and subsequent structural modifications. In Ref 16, the electronic reconstruction at interface was also included to describe metallic character. Assuming that the strain effect is relevant (as theoretically predicted in Ref 20), then one should also expect that a thinner film, preserving its strained state, will be also metallic. In other words, a conducting channel exists in LTO when grown on STO. Of course, this metallic character does not necessarily preclude an electronic interface reconstruction by electron transfer towards Ti-3d orbitals of STO. The answer of authors to this question is somehow circular and does not address the issue. Notice that the answer to this point is directly connected to points 1) & 2) above.

5) A question related to AMR, not previously disclosed, is intriguing. The authors report in Fig. 5 the AMR measurements (field in-plane). In general (with exceptions) AMR is observed when applying a magnetic field to a material either ferromagnetic or with magnetic impurities, at different directions with respect to the measuring current. The authors claim that SQUID data do not show indications of ferromagnetism (it is unclear what the authors want to express) and they also add that the AMR does not show hysteresis. They conclude that there is no long-range magnetic order. Notice that soft ferromagnets do not show hysteresis. As any magnetism in LTO/STO should be extremely soft, the conclusion of authors is incorrect. Therefore the above statements are not relevant or even misleading. Certainly, magnetic impurities (or magnetic pockets) may exist in 2DEGs and this could give rise to AMR.

Otherwise, which is the origin of AMR if magnetic scattering is absent?. Ref 51 is used to discuss the role of Rashba field in the context of AMR; however, in that reference I guess that magnetic scattering was assumed. It would have been useful to measure the magnetic susceptibility in two orthogonal in-plane directions to explore a possible contribution of

anisotropic Pauli susceptibility and thus different magnetization in to different orthogonal directions. In that case the discussion on AMR would be different.

The presence of magnetic moments at the interface, may be in the form of pockets as described above (or others), may produce features in the magnetic field dependent conductivity. It would have been useful to include somewhere in the manuscript or in the Suppl. Info. the temperature-dependent resistivity.

Overall, the manuscript reports interesting observations although in my view, it fails to provide a conclusive demonstration of the reported claims.

Responses to Reviewers' Comments:

REFEREE 3:

The revised version of the manuscript “Nontrivial Berry phase in quantum oscillations resulting from a giant Rashba spin splitting in LaTiO₃/SrTiO₃ heterostructures”, by M. J. Voit, et al., appears to have been improved with the introduction of significant changes. I agree with the present interpretation of the DFT results, and with the identification of the minority high mobility carriers with electrons filling dxz/dyz (hybridised) sub-bands, and of the majority low mobility carriers with electrons filling dxy sub-bands. By the way, this identification seems to have already been explicitly put forward in Ref. 38, if not even before.

We appreciate that referee #3 agrees with our present interpretation of the DFT results and the identification of the bands. We did not intend to give the impression that we are the first ones to suggest that the different mobility carriers come from different bands. We have now rephrased page 4 paragraph 5 so that it is clearer that others have proposed this before.

I am still dissatisfied with the discussion about the relation between the giant Rashba spin splitting and DFT calculations, but this comes rather to my fault, as I was not able to make myself understood in my previous report. I shall try to make my point clearer: given that the large electric field arising at the interface is captured by their DFT calculations, do the authors expect that any reasonable treatment of spin-orbit coupling within DFT may result in a giant Rashba-like spin splitting comparable to the one they deduce from the measurements? My wording referring to a field that must be “introduced by hand” was not meant to refer to the lack of polar discontinuity in the DFT results (I am sure that this effect is captured), rather to the possibility to quantitatively recover the giant Rashba-like spin splitting WITHIN the same DFT scheme, once (atomic) spin-orbit coupling is taken into account. My attitude is rather simple-minded here, but I am pretty sure the reader may have the same curiosity: if the authors had been able to obtain a reasonable estimate of the Rashba-like spin splitting within DFT, they would present their results in support of their data. The fact they did not, seems to indicate that this was not possible, and the reader may well wonder why. I ask the authors to discuss this aspect, which may be of interest to an even broader readership.

We thank the reviewer for clarification of his/her original question regarding whether a giant Rashba splitting appears in the DFT calculations when only the atomic spin-orbit coupling is incorporated. Unfortunately, we do not think that it is possible to get an accurate representation of a Rashba-type splitting because of the particular setup of the calculation. As described in the methods section, the chosen slab was LTO(3MLs)/STO(1.5MLs)/LTO(3MLs) (001): although the inversion symmetry breaking at the surface and interface is present on each side of the slab the overall slab is symmetric in order to avoid any spurious electric fields from periodic boundary conditions. However, Rashba-type splitting requires an inversion symmetry breaking. In particular, band 1 in Fig. 4 resides within the STO layer and due to the proximity of the two interfaces, it would not be possible to observe a Rashba-type splitting in this band with the DFT calculations. Also there may be issues with the thickness of the STO layer when attempting to determine a Rashba coupling from DFT calculations (see for example Kosugi et al (J. Phys. Soc. Jpn. 80, 074713 (2011)) for the dependence of the Rashba effect in Au on the thickness of the slab).

We have indeed performed DFT with and without atomic spin orbit coupling. When comparing these results, we do find some Rashba-like splitting near the Γ point of some bands and some splitting at band crossings. However, such splitting is not as large as the splitting deduced from our experiments. In order to determine the details of the band structure around the Γ point at higher resolution, we have performed tight binding calculations. These calculations, combined with the effective masses from our quantum oscillation experiments, are consistent with Rashba coupling constant deduced from the weak anti-localization correction to the magnetoconductivity.

We now include a discussion to this effect in the manuscript in the methods section on the DFT calculations.

Besides this clarification, I think that, although not fully conclusive as far as the identification of the relevant electron states is concerned, the present work is rather interesting and sound, and will certainly trigger new experimental and theoretical investigation on oxide heterostructures.

We appreciate the fact that the referee “[thinks] *the present work is rather interesting and sound, and will certainly trigger new experimental and theoretical investigation on oxide heterostructures*” and that he/she recommends that “*their piece of work may be accepted for publication in Nature Communications ... once the authors have considered [his/her] suggestions.*”

REFEREE #4:

I agree with Reviewer #1 that there is some inconsistency between the Hall data and the Shubnikov-de Haas measurements as I mentioned in my first review. The π Berry phase is indeed interesting assuming that the error on this phase is estimated correctly. I would leave these questions to the readers of the paper. I recommend publication.

We thank referee #4 for his/her recommendation for publication. He/she suggests that some questions be left up the reader. However, we did revisit the “*inconsistency between the Hall data and the Shubnikov-de Haas measurements as [the referee] mentioned in [his/her] first review.*”

For reference referee #4 described in his/her original feedback that “*from the Hall and SdH analysis the authors obtain three bands: The first observed in low field SdH with 10^{11} , another one with 10^{12} both detected by the SdH as well and a third band that accommodates the majority of the carriers, which is not seen by SdH but dominates the Hall. How come the 10^{12} band is not observed in the Hall and the 10^{11} band is? The authors should clarify this point.*”

In our most recent revision, we believe that we had articulated how the two types of carrier concentrations deduced from the Hall effect measurements *do not* map onto the two sets of quantum oscillations. Quantum oscillations require carriers with high enough mobility. It would be impossible to reliably fit a three-band model to the Hall effect with the two high

mobility carriers from the SdH oscillations and another low mobility carrier. So, we instead assume similarly high mobilities for the two carriers observed by SdH oscillations. With this assumption, we can simplify the analysis of the Hall effect data to an effective two carrier model with one carrier being the low mobility carriers not observed by SdH oscillations and the other “carrier” being the sum of the high mobility carriers observed by SdH oscillations.

In our case here, the sum of the carrier concentrations from the two sets of quantum oscillations ($1.7 \times 10^{12} \text{ cm}^{-2} + 1.8 \times 10^{11} \text{ cm}^{-2}$) is consistent with the effective low carrier concentration/high mobility deduced from the Hall effect ($2.0 \times 10^{12} \text{ cm}^{-2}$). The high carrier concentration/low mobility carriers deduced from the Hall effect represent a separate band. So, there is no inconsistency between the Hall data and the SdH measurements. However, we clearly did not articulate this explanation well enough. We did not mean to suggest that one of the bands is not measured in the Hall effect as the referee says.

Therefore, we have rewritten page 4 paragraph 4 to more clearly articulate the correlation between the carriers measured in the Hall effect data and those measured by Shubnikov de Haas oscillations.

REFEREE #5:

The manuscript by M.J. Veit et al. deals with the observation of a large Rashba-type spin splitting at the interface of thin LaTiO3 films on SrTiO3 by magnetotransport experiments. I have read the full manuscript file and all previous referee comments and the response of the authors to those comments.

The authors have performed a systematic study and give enough background information to check the validity of their claims. I also appreciate that they tone down their claims and use language along the lines that the results are “consistent with a large Rashba splitting”. Indeed all the observations are consistent with such an interpretation, but don’t provide a definitive proof. As a result, reviewer #1 draws into question the physical interpretation of the results and suggests that there might be another explanation. The main reason for doubting the interpretation is that the found Rashba-type spin splitting should be “extraordinary large” and the “largest one ever reported in oxides.” Also reviewer #3 is surprised by the magnitude of the found Rashba splitting, and the authors themselves also state that the splitting is “surprising”. It appears to me that reviewer #1 would be willing to accept the results if the found splitting appears to be reasonable.

In my opinion the reason for this surprise about the magnitude of the Rashba-type spin splitting is due to the rather simplistic model used by both authors and reviewers with regard to the origin of the Rashba-type spin splitting. In this model a electric field is responsible for the size of the spin splitting, which is of course the original interpretation for semiconductor heterostructures. However, the study of Rashba-type effects at surfaces and also bulk systems has lead the field to the understanding that such fields play only a very minor role and that the main contributions are of a structural nature. A buckled atomic structure will result in a extreme enhancement of the spin splitting. In J. Phys.: Condens. Matter 21 (2009) 403001 a review of this argumentation can be found.

The polar nature of STO and LTO in combination with relaxation at the interface can result in a large atomic corrugation and thus Rashba effect. For the STO to vacuum interface this results in

a Rashba constant of 5×10^{-11} eVÅ (Nature Materials, 13, 1085 (2014)), more than 2 times as large as found here. These results are still being debated and the spin splitting found in the present manuscript can help shed a light on this. In a general sense one can thus state that the found Rashba-type spin splitting is not surprisingly large, but along the lines of what can be expected for a strongly localized 2D state at a buckled interface. Inclusion of a discussion along these lines will enhance the quality of the manuscript even further and should also convince reviewer #1 that the Rashba splitting is indeed the simplest explanation, also given the Berry phase.

We thank the referee for pointing out that by including the effect of a buckled interface, the large magnitude of the Rashba splitting is not as large compared to previous studies. We have now included the cited reference in the reference section as well as a discussion of the role of structure describing how indeed the Rashba splitting is the simplest explanation on page 6 paragraph 2.

In my opinion the manuscript is suitable for Nature Communications and can be accepted for publication if the following changes are made:

We appreciate that referee #5 believes that the manuscript is suitable for *Nature Communications* and can be accepted for publication if certain changes are made. We address below each of the requested changes.

- 1. The authors should remove the claim of the largest Rashba-type spin splitting in oxides and include a reference and discussion of the Nature Materials paper mentioned above.*

As requested, we have now removed the claim that we have a surprisingly large spin orbit splitting and include the *Nature Materials* reference. We add a discussion of a Rashba-type spin splitting in the context of this Nature materials paper describing the large spin splitting observed in the 2DEG at the surface of SrTiO₃ crystals. In this discussion, we compare the magnitude of the Rashba-type spin splitting in our LaTiO₃/SrTiO₃ sample to that of the surface 2DEG of SrTiO₃ on page 4 paragraph 2.

- 2. The authors should revise their discussion of the Rashba effect to include structural effects. Furthermore, under these considerations it is technically no longer correct to refer to it as Rashba splitting, but typically one uses Rashba-type spin splitting in literature.*

As requested, we have now revised our discussion of the Rashba effect to include structural effects on page 6 paragraph 2. This includes a mention of our own DFT calculations which indeed show evidence for buckling at the interface. We have also replaced reference to “Rashba splitting” to “Rashba-type spin splitting” in various places in the manuscript.

- 3. The discussion and use of terminology of the effective mass is rather confusing. In a first approximation, the Rashba effect should not influence the effective mass when referring to the curvature. In line 244 the authors make the link to the effective mass as determined from their*

experiments and I agree with this statement, but the statement in line 205 can be misinterpreted and should be rephrased for clarity.

We thank the referee for pointing out the possible confusion that may arise from our discussion of the effective mass. We now have rephrased the statement on page 3 paragraph 5 to include the terminology “cyclotron effective mass” as we use in line 244 for clarity.

- 4. It would be helpful if the authors show a band structure around Gamma as obtained from their tight-binding analysis including the correct energy and momentum scale. This will make it easier to compare the results to other studies.*

As requested we now include a plot of the band structure around the gamma point obtained from the tight binding analysis along with the DFT results in the same plot for comparison in supplementary section 5.

- 5. The states in the seminal ref. 33 actually are the surface states of BiTeI, as was found out in later studies (PRL 109, 116403 (2012) and PRL 109, 096803 (2012)). The bulk Rashba constant was determined in PRL 109, 116403 (2012).*

We thank the referee for pointing this out and have replaced reference 33 with the PRL reference pointed out by the referee.

REFEREE #6:

After revising the manuscript and the correspondence with Referees, I'm commenting in the following the main aspects that, in my opinion, remain disputed.

1) Referee #1 argued that bands arising from LTO and STO rather than Rashba-induced band splitting could be relevant for the observed two different oscillations. In their answer, the authors disagree with the referee opinion and included a new plot (Fig 4) intended to convince referee (and potential readers) that the xy band (labelled 2) and zz+yz bands (labelled 1) crossing the Fermi level, are of STO parentage. Fig. 4 only shows that a band (labelled 2) crosses the Fermi level, and a band (labelled 1) slightly touches the Fermi level. Unfortunately, this plot does not allow visualizing easily the parentage of the corresponding orbitals. This is better seen in Figs. S6 and S7 in Supplementary information. Whereas Fig S6 shows that interfacial xy orbital dominate the Fermi level, a smaller contribution of xz+yz of neighboring STO and LNO layers also exist (Fig S7). Although the STO-derived density of states is somewhat higher than that of LTO the difference seems to be rather marginal. If so, STO and LTO carriers will participate to conduction, probably with different effective masses and mobilities. It seems that authors do not appear to consider the contribution of LTO subbands. This is somewhat surprising as authors are aware (Ref 16) that ultrathin (strained) LTO films on STO are metallic.

As the referee points out, we have included three plots (Figure 4, S6 and S7) delineating the density of states as derived from the DFT calculations. The referee states that the “[Figure 4] does not allow visualizing easily the parentage of the corresponding orbitals.” However, the point of Figure 4 was to show an overview of the DFT results with bands from LaTiO₃, SrTiO₃ as well as the interface. We had chosen to focus the reader’s attention

in Figure 4 on those bands that come close to the Fermi level (blue and orange) but also show bands far above and below the Fermi level that should not affect transport. Of the bands close to the Fermi level, DFT attributes none of them to the LTO layers which correspond to +1IF, +2IF and +3IF in Figure S6 and S7. This means that at least the first three-unit cells of LaTiO_3 do not appear to contribute to the metallicity observed in our ultra-thin films.

The referee says that “*although the STO-derived density of states is somewhat higher than that of LTO the difference is rather marginal.*” Actually, there is NO density of states at the Fermi level from any LTO layer (either IF+1, IF+2 or IF+3) in Figures S6 and S7. We believe this confusion comes from our color scheme and labeling in the previous figure. In order to make this absolutely clear in Figures S6 and S7, we have changed the colors used for the different layers and changed the labeling of the LaTiO_3 layers from +1IF, +2IF, +3IF to 1LTO, 2LTO and 3LTO and -1IF to STO. in supplementary section 4.

We would like to point out that Reference 16 (to which the referee refers for ultra-thin strained LTO films which are metallic and is work from our group) only describes films from 15nm-85nm. These results are in a thickness regime that cannot be directly compared with the samples in the current study which is why we used the terminology “ultra-thin films” to distinguish the current results from our previous studies. Resistivity values in 15-85nm thick films in Ref 16 are basically the same. In fact, we have found that the metallic resistance in strained LTO film scales down to 2.5nm. Below this thickness, the resistivity of the films deviates from that found in thicker LTO films (see Figure R1 below). The deviation indicates that there is a different conductance channel that appears in this ultra-thin thickness regime and that perhaps can be attributed to electronic reconstruction. It is only in this ultra-thin thickness regime that we observe quantum oscillations in the magnetoresistance.

Therefore, the fact that DFT does not indicate that there are LTO sub bands crossing the Fermi level in 3-unit cell thick films and that experimentally the resistivity behavior of 3-4-unit cell thick LTO films is different from the behavior of thicker films is consistent with *the picture that metallic conduction in 3-4-unit cell thick LTO films is dominated by an interface conduction channel whose origin may be found in electronic reconstruction.*

In order to make this clearer, we now have added an additional supplementary section explaining and showing a plot of the temperature dependent resistivity (as the referee requested) for LTO films from 60nm down to 1.2nm. In this section, we now describe how there is a separate conduction regime for LTO films below 2.5nm where quantum oscillations emerge in the magnetoresistance and that this is NOT at odds with the bulk metallicity observed in LTO films 15-85nm thick in reference 16.

2) Carrier concentration and mobility are extracted from Hall data and used to support the claim that there are two sources of carriers. This method is much used but results are not always reliable. Basically, the problem arises because the Hall voltage is a very smooth function and the extracted parameters are much correlated. In fact, inspecting the extracted values for each transport channel, it turns out the corresponding carrier density and mobility values are different but,

suspiciously, their product (carrier density \times mobility) ($\approx 8-9 \cdot 10^{15}$) is almost identical for both channels. In this context, it is interesting to note that in Supplementary Infor (III) where the fitting of Hall data is described, the values of the carrier density and mobility for each band are different than those in the body of the manuscript. This may illustrate the fact that the fitting is not unique and thus the extracted values and their claimed agreement with data obtained from ShH oscillations may not be fully significant.

As is acknowledged by those in the field, the two-carrier fit of the Hall resistance is not perfectly reliable. However, this technique does give a basic picture and is often used in conjunction with other methods of extracting the carrier concentration, such as Shubnikov de Haas oscillations, to develop a more complete picture of the carrier concentration distribution in materials. However, the carrier concentration values deduced from the Shubnikov de Haas oscillations cannot be disputed. The sum of the carrier concentration values from the two pockets corresponding to the low field and high field oscillations match the carrier concentration value of the high mobility carriers derived from the Hall voltage.

Given the fact that we complement the carrier concentration values deduced from the Hall effect with those from Shubnikov de Haas oscillations, we believe that we have performed an accurate analysis as possible and that the analysis is certainly standard in the literature.

We apologize for the typos showing different values for the carrier concentration and mobility values in the main text and the supplementary section. The values in the main text are correct, and we have now fixed these typos in section 3 in the supplementary material.

3) A value of the Berry phase is deduced from the observed oscillations of $R(H)$. The relevance of the extracted value, assuming it is meaningful, is not much discussed. However, it is emphasized in a rather mysterious way (“nontrivial Berry phase ...”) in the title. I’m not sure that the title is indicative of the strength of the manuscript. In fact, it seems that the authors only use the extracted value (π) as a potential support to the presence of Rashba splitting.

We would like to point out that we have described the importance and significance of the value of the derived Berry’s phase in an entire paragraph (page 5 paragraph 5). In this paragraph, we go into great detail as to how a “nonzero Berry’s phase may be realized for charge carriers that have an orbit around the Dirac point” and that we would expect this non-zero Berry’s phase to be π , providing relevant references.

In order to make sure that the reader does not skip over this paragraph, we now have rewritten the first sentence of this paragraph to indicate explicitly that this paragraph will explain the significance of a Berry’s phase of π . We have also added a few sentences on the general significance of a nonzero Berry’s phase into this paragraph.

In addition, we have also changed the title of the paper from “nontrivial” to “nonzero” to make the significance of the Berry’s phase less “mysterious”. We would like to note that there are only a handful of explanations for a Berry’s phase of π . We go through these possibilities in the discussion section on page 6 paragraph 2. Given the other possible

explanations, the one that we propose of charge carriers having an orbit around a Dirac point associated with Rashba splitting is supported by our other experimental measurements – including weak anti-localization correction to the magnetoconductivity, anisotropic magnetoresistance, and two sets of quantum oscillations at the Fermi level.

4) Referee #2 raises a crucial question regarding material's issue. The point is that LTO thin films (< 10 uc) are metallic if grown on STO (ref 15 and 16, by authors). In reference 15 it was first argued that metallicity comes from strain effects and subsequent structural modifications. In Ref 16, the electronic reconstruction at interface was also included to describe metallic character. Assuming that the strain effect is relevant (as theoretically predicted in Ref 20), then one should also expect that a thinner film, preserving its strained state, will be also metallic. In other words, a conducting channel exists in LTO when grown on STO. Of course, this metallic character does not necessarily preclude an electronic interface reconstruction by electron transfer towards Ti-3d orbitals of STO. The answer of authors to this question is somehow circular and does not address the issue. Notice that the answer to this point is directly connected to points 1) & 2) above.

As we have described in our response to point #1, the origin of the metallicity in thicker LaTiO₃ films is different from that of ultrathin films (by which we mean 3-4-unit cells). Our own work (reference 16) shows that for films 15-85nm thick, the resistivity is the same. Indeed, we have extended this study down to 1.2nm or 3-unit cells and found that down to 2.5nm, the resistivity of these films exhibit the same metallic resistivity values. However, for films 3 and 4-unit cells thick (1.2-1.5nm) the resistivity deviates from that of thicker films. We now include a supplementary section which shows the temperature dependence of the resistivity to illustrate this. This deviation indicates that there is a different or additional source of metallic conduction in ultra-thin (3-4-unit cell thick) LaTiO₃ films.

Our DFT calculations suggest that there are no carriers originating from the LaTiO₃ layer (from any of its sub bands) in the 3-unit cell LaTiO₃ film on SrTiO₃. In these films, according to DFT calculations, carriers appear to come from the interface layer and the SrTiO₃ layer. Therefore, DFT calculations would be consistent with a source of metallic conduction at the interface different from the bulk metallic states of compressively strained LaTiO₃ films. This interface conduction may be attributed to some sort of electronic reconstruction at the interface.

Evidence for interface conduction from our experiments on ultra-thin films and these DFT calculations plus the results of reference 20 indicative of bulk metallicity in thicker films suggest that there are two thickness regimes where different conduction mechanisms dominate the measurements. In thick films, a bulk conduction dominates a resistance or Hall effect measurement due to size effects. Because these oxide materials can remain strained for 100s of nanometers, the strain induced bulk conductivity suggested by reference 20 is measured in the films in references 15 and 16. Note that this does not preclude an electronic reconstruction in these thick films. Basically, we cannot measure the contribution from electronic reconstruction due to the small resistance of the bulk channel compared to the larger resistance from the electronic reconstruction channel confined to the interface.

Theoretically, our current DFT calculations suggest that the strain induced conduction is suppressed by an electronic reconstruction at the interface for ultra-thin films. We note that reference 20 which combined dynamical mean field theory and DFT to predict metallicity of the bulk material due to strain induced electronic structure modification. By combining the two models of ultrathin films and bulk, we begin to assemble a complete picture of the transport behavior from the ultra-thin limit to thicker LaTiO₃ films on SrTiO₃.

We now clearly describe the different conduction mechanism for thicker LaTiO₃ films where the bulk of the thin film is metallic versus ultra-thin (3-4-unit cell) LaTiO₃ films where the interface is primarily responsible for metallicity in page 2 paragraph 2. We believe that we now clearly articulate why our current results are consistent with our own previous results in reference 16.

5) A question related to AMR, not previously disclosed, is intriguing. The authors report in Fig. 5 the AMR measurements (field in-plane). In general (with exceptions) AMR is observed when applying a magnetic field to a material either ferromagnetic or with magnetic impurities, at different directions with respect to the measuring current. The authors claim that SQUID data do not show indications of ferromagnetism (it is unclear what the authors want to express) and they also add that the AMR does not show hysteresis. They conclude that there is no long-range magnetic order. Notice that soft ferromagnets do not show hysteresis. As any magnetism in LTO/STO should be extremely soft, the conclusion of authors is incorrect. Therefore the above statements are not relevant or even misleading. Certainly, magnetic impurities (or magnetic pockets) may exist in 2DEGs and this could give rise to AMR.

Otherwise, which is the origin of AMR if magnetic scattering is absent?. Ref 51 is used to discuss the role of Rashba field in the context of AMR; however, in that reference I guess that magnetic scattering was assumed. It would have been useful to measure the magnetic susceptibility in two orthogonal in-plane directions to explore a possible contribution of anisotropic Pauli susceptibility and thus different magnetization in to different orthogonal directions. In that case the discussion on AMR would be different.

The presence of magnetic moments at the interface, may be in the form of pockets as described above (or others), may produce features in the magnetic field dependent conductivity. It would have been useful to include somewhere in the manuscript or in the Suppl. Info. the temperature-dependent resistivity.

Overall, the manuscript reports interesting observations although in my view, it fails to provide a conclusive demonstration of the reported claims.

We have performed bulk SQUID magnetometry measurements as well as element specific x-ray magnetic circular dichroism measurements. First of all, we find that the only magnetic signal that we observe in our magnetization versus applied magnetic field scans is a diamagnetic signal from the SrTiO₃ substrate. A closer look at the temperature dependence of the magnetization shows a paramagnetic tail at low temperatures which is found in the substrates before deposition. Susceptibility measurements of the samples show

a strong diamagnetic signal from room temperature down to 2K with no trace of any ferromagnetism regardless of what direction the field is applied.

We have also performed x-ray magnetic circular dichroism (XMCD) measurements at 10 K which can measure element specific magnetic signal. We have extensive experience performing these experiments at the Advanced Light Source and say with confidence that we do not observe any magnetic signal from the Ti as well as any trace magnetic signal from random other 3d transition metals such as Fe and Co. These measurements show clearly that we observe neither ferromagnetic ordering nor magnetic impurities.

We would also like to point out that the referee's statement that "soft ferromagnets do not show hysteresis" needs to be qualified and is not a generally correct statement. In a single crystalline ferromagnetic sample, we would expect that magnetization reversal occurs via domain wall motion, coherent rotation of moments or a combination of both. Depending on the direction of the applied magnetic field, we may observe reversible or hysteretic behavior or a combination of the two. For a soft ferromagnet, the magnetic anisotropy is small and hence the coercivity is usually small, but when the magnetic field is applied in a magnetically easy direction, the coercive field is nonzero and measurable. Even in the case of single crystal yttrium iron garnet which is extremely soft, a small but finite coercive field is measurable. When the field is applied along a magnetically hard direction (due to crystalline, shape or other effects), the sample may show reversible behavior without any coercive field. None of our magnetic field dependent measurements (magnetoconductivity in Fig. 1, Hall effect in Fig. 1, SQUID, or AMR in Fig. 5 and Fig. S9) show any hysteresis down to an ~ 1 Oe resolution, and there are no abrupt features around zero field in any measurement that we would associate with a ferromagnet switching directions.

AMR can come from ferromagnetism or ferromagnetic impurities but these are not the only sources of AMR. We had included reference 52 which goes into detail on how AMR can arise from a two-dimensional Rashba system with nonmagnetic disorder. This paper also explains the peak in the field dependence of the AMR (shown in supplementary figure S9). Given that we do not observe any evidence of ferromagnetism or magnetic impurities, this interpretation seems the most likely.

We now explicitly state that we do not observe any hysteresis in the field dependence of the magnetization of the samples in SQUID magnetometry or magnetic signal above the noise level in XMCD on page 6 paragraph 1 to make clear that we do not find any evidence for magnetic impurities in our samples.

Finally, the referee states that we should include "temperature dependent resistivity." We have now included a plot of this in supplementary section 8 along with the temperature dependent resistivity of thicker LaTiO_3 films.

We list in detail below the list of major modifications to the manuscript.

Major Changes to the Manuscript

1. Changed title to “*Nonzero Berry Phase in Quantum Oscillations Resulting from a Giant Rashba Spin Splitting in LaTiO₃/SrTiO₃ Heterostructures*”
2. Removed the statement of this being the largest observed Rashba effect in oxides.
3. Added the following to page 2 paragraph 2: “*Additionally, the ultra-thin samples show a significantly higher resistivity than the thicker samples (see Supplementary Material), indicating that there are two thickness regimes where different conduction mechanisms dominate. In the thick ($>2\text{nm}$) regime, tetragonal distortions modify the electronic structure giving rise to metallicity in the bulk of the LTO. However, for ultra-thin (3-4-unit cells) films electronic reconstruction at the interface dominates conduction. The oscillations are therefore a measurement of the electronically reconstructed interface.*”
4. Rephrased the discussion of the effective mass on page 3 paragraph 4 to include the use of the term “cyclotron mass”: “*This value is considerably larger than the measured cyclotron effective mass (defined as the derivative of the Fermi Surface area). So, we cannot attribute either of the oscillations to pure d_{xz} or d_{yz} bands. Moreover, both of the measured cyclotron masses are different from the calculated d_{xy} and hybridized bands. However, a Rashba effect would create an inner band with a smaller cyclotron mass and an outer band with a larger cyclotron mass than the unsplit band. So, the observed effective mass values are possibly consistent with quantum oscillations arising from a Rashba split d_{xy} band or d_{xz+yz} hybridized band.*”
5. Added the following sentence to page 4 paragraph 1: “*A similar but larger constant of $5 \cdot 10^{-11} \text{ eVm}$ has been found at the surface of cleaved STO.*”
6. Added page 4 paragraph 3: “*Given that neither of the carrier concentrations derived from the SdH oscillations frequencies match the high concentration carriers from the Hall effect fit, additional carrier types and hence a three-band model may be appropriate. However, such a problem is not constrained enough to produce accurate or meaningful values. A three-band model is easily simplified to a two-band model if we assume that two of the bands have the same mobility. The model then becomes a two-band model with one of the “carriers” having the concentration of the sum of the two similar mobility bands. In our case, the sum of the sheet concentrations deduced from the oscillations is equal to the low carrier concentration associated with high mobility carriers deduced from the Hall effect, thus indicating that the oscillations account for all of the high-mobility/low-concentration carriers.*”
7. Rephrased page 5 paragraph 5 to include a general discussion of the Berry phase: “*In general, a nonzero Berry phase is the result of a band crossing. For the case of a perfectly linear Dirac point, the Berry phase is π but is reduced and vanishes as a gap is introduced. For systems that are described by a Rashba Hamiltonian, the point where the spin-split energy bands cross should correspond to a Dirac point. As a result, a Berry phase of π may be realized for charge carriers that have an orbit around the Dirac point. Our extracted value of π therefore indicates a Rashba-type splitting induced band crossing with a small, if any, gap. Well separated oscillations and a nonzero Berry's phase have been observed in the bulk Rashba semiconductor BiTeI. However, in typical Rashba systems, the two frequencies are similar so the result is a beating pattern which prevents an accurate*

extraction of the Berry's phase. To date, there has been no report of such Berry's phase in complex oxide interface systems”

8. Added the following sentence to page 6 paragraph 1: *“Additionally, XMCD measurements at 10 K show no signs of magnetism or common magnetic impurities such as Fe or Co.”*
9. Added the following sentence to page 6 paragraph 2: *“Additionally, structural effects such as buckling at the interface can greatly enhance such a Rashba-type effect. Our DFT results show evidence for a possible buckling near the interface in ultra-thin films. So, it is not surprising that we observe such large Rashba-type effects.”*
10. Added the following sentence to the DFT portion of the Methods section: *“Although at each interface inversion symmetry is broken, the symmetry of the overall system and possible interference between the two interfaces does not allow us to properly model Rashba-type splitting. This aspect of the DFT calculations goes beyond the present study.”*
11. Modified the color scheme and labeling of Figure S6 and S7 to make clear the origin of the density of states from each layer in supplementary section IV.
12. Added supplementary section V to compare DFT and tight binding calculations in Figure S8: *“Figure S8 shows a close up of the tight binding model described in the main text along with the hybridized d_{xz+yz} band 1 from the DFT calculations along the Γ -X and Γ -M directions. As described in the main text, the tight binding hopping parameters were first adjusted to fit band 1 from the DFT calculations (light blue line in Fig. S8) which produced the dashed black line in Fig. S8. A linear Rashba term was then added to fit the area and cyclotron mass from the SdH oscillations to produce the red and blue curves.”*
13. Added supplementary section VIII showing the temperature dependent resistivity for a series of LaTiO₃ films on SrTiO₃ in Figure S11: *“Figure S11 shows the temperature dependence of the resistivity for LTO films of a wide range of thicknesses. As seen in this figure, the resistivity is thickness independent for films thicker than 2 nm, but increases dramatically when the film is only a few unit cells thick. Additionally, the ultra-thin films have a flatter temperature dependence than the thicker films. This indicates that there is likely a different source of conduction dominating the electrical transport in these ultra-thin films than the thick films.”*

Reviewers' Comments:

Reviewer #3 (Remarks to the Author):

The manuscript “Nonzero Berry phase in quantum oscillations resulting from giant Rashba-type spin splitting in LaTiO₃/SrTiO₃ heterostructures”, by M. J. Veit, et al., is undeniably controversial. After examining the rather lengthy correspondence with several referees, it appears that the issues raised at various stages of the review process are all meaningful. The authors' replies are reasonable, but leave many questions open and raise further questions. My personal opinion is that the results presented in this piece of work are not fully conclusive, but are novel and interesting enough, and may foster future investigation. So, the question is whether the scales is slightly tilted in favour or against publication. At the present stage, my suggestion is that the manuscript should be accepted for publication in Nature Communications, and be opened to debate within the scientific community.

Reviewer #4 (Remarks to the Author):

The paper has improved further in the last round of reviews. The authors have satisfactorily responded to all Referees comments and criticisms. In particular, I find the response to all points raised by Referee #6 reasonable.

A final comment the authors may want to consider: The authors discuss the contribution of the various Ti d-bands to the conductivity and to the quantum oscillations. In fact, when (atomic) spin-orbit interaction is present the bands cannot be described as "pure" d_{xy} or d_{yz} etc. but they are mixed together. This has been shown by Joshua et al. doi:10.1038/ncomms2116 and later extended by Maniv et al. doi:10.1038/ncomms9239 to explain both the quantum oscillations and the Hall conductivity.

I would be happy to see these interesting results published in Nature Communications.

Reviewer #5 (Remarks to the Author):

The authors have provided extensive and convincing responses to all the questions and have made the relevant changes to the manuscript. I now recommend the manuscript for publication in Nature Comm.

Reviewer #6 (Remarks to the Author):

The authors have addressed and explained the major questions from all Referees. In my opinion, the manuscript is now suitable for publication.

Responses to Reviewers' Comments:

Referee 3:

The manuscript "Nonzero Berry phase in quantum oscillations resulting from giant Rashba-type spin splitting in LaTiO₃/SrTiO₃ heterostructures", by M. J. Veit, et al., is undeniably controversial. After examining the rather lengthy correspondence with several referees, it appears that the issues raised at various stages of the review process are all meaningful. The authors' replies are reasonable, but leave many questions open and raise further questions. My personal opinion is that the results presented in this piece of work are not fully conclusive, but are novel and interesting enough, and may foster future investigation. So, the question is whether the scales is slightly tilted in favour or against publication. At the present stage, my suggestion is that the manuscript should be accepted for publication in Nature Communications, and be opened to debate within the scientific community.

We thank the referee for his/her time evaluating our work and for his/her recommendation for publication.

Referee 4:

The paper has improved further in the last round of reviews. The authors have satisfactorily responded to all Referees comments and criticisms. In particular, I find the response to all points raised by Referee #6 reasonable.

A final comment the authors may want to consider: The authors discuss the contribution of the various Ti d-bands to the conductivity and to the quantum oscillations. In fact, when (atomic) spin-orbit interaction is present the bands cannot be described as "pure" d_{xy} or d_{yz} etc. but they are mixed together. This has been shown by Joshua et al. doi:10.1038/ncomms2116 and later extended by Maniv et al. doi:10.1038/ncomms9239 to explain both the quantum oscillations and the Hall conductivity.

We thank referee #4 for bringing these papers to our attention. Our DFT calculations certainly indicate that when atomic spin-orbit coupling is added, there is additional hybridization of the t_{2g} orbitals. This hybridization does not greatly modify the electronic structure near the Fermi level but is important to note. So, we have now added a sentence to page 3 paragraph 3 and included the two references.

Referee 5:

The authors have provided extensive and convincing responses to all the questions and have made the relevant changes to the manuscript. I now recommend the manuscript for publication in Nature Comm.

We thank the referee for all the previous comments and for the recommendation for publication.

Referee 6:

The authors have addressed and explained the major questions from all Referees. In my opinion, the manuscript is now suitable for publication.

We thank the referee for his/her constructive comments and for the recommendation for publication.